# Autoregressive Models in Vision: A Survey

**Jing Xiong**[1,†]    **Gongye Liu**[2,†]    **Lun Huang**[3,10]    **Chengyue Wu**[1]    **Taiqiang Wu**[1]
**Yao Mu**[1]    **Yuan Yao**[4]    **Hui Shen**[5]    **Zhongwei Wan**[5]    **Jinfa Huang**[4]
**Chaofan Tao**[1,‡]    **Shen Yan**[6]    **Huaxiu Yao**[7]    **Lingpeng Kong**[1]    **Hongxia Yang**[9]
**Mi Zhang**[5]    **Guillermo Sapiro**[8,10]    **Jiebo Luo**[4]    **Ping Luo**[1]    **Ngai Wong**[1]

[†] **Equal Contributors.** [‡] **Corresponding Author. Email:** `cftao@connect.hku.hk`

[1]**The University of Hong Kong**    [2]**Tsinghua University**    [3]**Duke University**
[4]**University of Rochester**    [5]**The Ohio State University**    [6]**Bytedance**
[7]**The University of North Carolina at Chapel Hill**    [8]**Apple**
[9]**The Hong Kong Polytechnic University**    [10]**Princeton University**

**Reviewed on OpenReview:** `https://openreview.net/forum?id=1BqXkjNEGP`

## Abstract

Autoregressive modeling has been a huge success in the field of natural language processing (NLP). Recently, autoregressive models have emerged as a significant area of focus in computer vision, where they excel in producing high-quality visual content. Autoregressive models in NLP typically operate on subword tokens. However, the representation strategy in computer vision can vary in different levels, *i.e.*, pixel-level, token-level, or scale-level, reflecting the diverse and hierarchical nature of visual data compared to the sequential structure of language. This survey comprehensively examines the literature on autoregressive models applied to vision. To improve readability for researchers from diverse research backgrounds, we start with preliminary sequence representation and modeling in vision. Next, we divide the fundamental frameworks of visual autoregressive models into three general sub-categories, including pixel-based, token-based, and scale-based models based on the representation strategy. We then explore the interconnections between autoregressive models and other generative models. Furthermore, we present a multifaceted categorization of autoregressive models in computer vision, including image generation, video generation, 3D generation, and multimodal generation. We also elaborate on their applications in diverse domains, including emerging domains such as embodied AI and 3D medical AI, with about 250 related references. Finally, we highlight the current challenges to autoregressive models in vision with suggestions about potential research directions. We have also set up a Github repository to organize the papers included in this survey at: `https://github.com/ChaofanTao/Autoregressive-Models-in-Vision-Survey`.

## 1  Introduction

Autoregressive models, which generate data by predicting each element in a sequence based on the previous elements through conditional probabilities, initially gained prominence in the field of natural language processing (NLP) (Vaswani, 2017; Radford et al., 2019; Brown et al., 2020; Achiam et al., 2023; Wan et al., 2023; Zhou et al., 2023a). This success can be attributed to their inherent advantage of capturing long-range dependencies and producing high-quality, contextually relevant outputs. Especially empirical scaling laws (Henighan et al., 2020; Hoffmann et al., 2022; Muennighoff et al.,

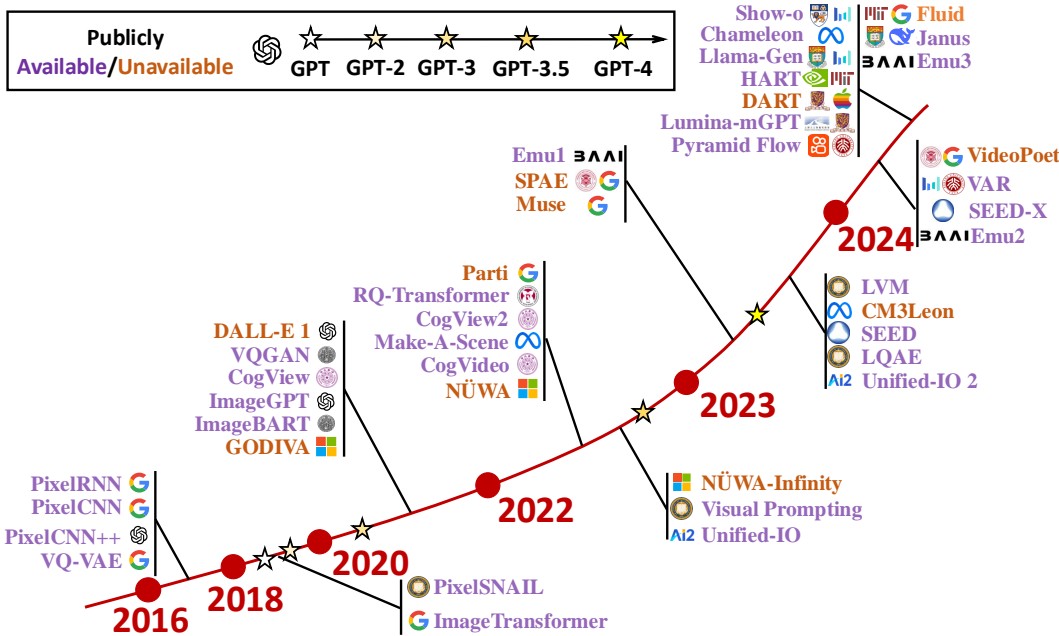

Figure 1: We provide a timeline of representative visual autoregressive models, which illustrates the rapid evolution of visual autoregressive models from early pixel-based approaches like PixelRNN in 2016 to various advanced systems recently. We are excitedly witnessing the rapid growth in this field.

2023; Tao et al., 2024; Lyu et al., 2023) reveal that increasing model size and compute budgets consistently improves cross-entropy loss across various domains like image generation, video modeling, multimodal tasks, and mathematical problem solving, following a universal power-law relationship. Inspired by their achievements in NLP, autoregressive models have recently begun to demonstrate formidable potential in computer vision.

The timeline in Figure 1 illustrates the key milestones and developments in the evolution of visual autoregressive models, highlighting their transition from NLP to computer vision. To date, autoregressive models have been applied to a wide array of generative tasks, including image generation (Parmar et al., 2018; Chen et al., 2020), image super-resolution (Guo et al., 2022; Li et al., 2016), image editing (Yao et al., 2022; Crowson et al., 2022), image-to-image translation (Li et al., 2024e;d) and video generation (Tulyakov et al., 2018; Hong et al., 2022), multi-modal tasks (Yu et al., 2023c; Lu et al., 2022) and medical tasks (Ren et al., 2024a; Tudosiu et al., 2024), *etc.* This broad applicability underscores the potential for further exploration and application of autoregressive models.

With the rapid proliferation of visual autoregressive models, keeping up with the latest advancements has become increasingly challenging. Therefore, a comprehensive survey of existing works is both timely and crucial for the research community. This paper endeavors to provide a thorough overview of recent developments in visual autoregressive and explores potential directions for future improvements.

We emphasize that there are at least three distinct categories of visual autoregressive models defined by their sequence representation strategies: pixel-based, token-based, and scale-based models. Pixel-RNN (Van Den Oord et al., 2016), as a representative pixel-wise model in the pioneering of next-pixel prediction by transforming a 2D image into a 1D pixel sequence, capturing both local and long-range dependencies but with high computational cost. Next-token prediction, inspired by NLP, compresses images into discrete tokens for efficient high-resolution processing, exemplified by models like VQ-VAE (Van Den Oord et al., 2017). VAR (Tian et al., 2024) introduces next-scale prediction, a hierarchical method that generates content across multiple scales, from coarse to fine autoregressively

capturing visual information at multiple resolutions. Each category offers unique advantages and challenges, making them promising directions for future research.

We further introduce a multi-perspective categorization of autoregressive models applied to computer vision, which classifies existing models based on criteria such as the sequence representation strategy, the underlying framework, or the target task. Our categorization aims to provide a structured overview of how these models are utilized across various vision tasks. We then present both quantitative and qualitative metrics to assess their performance and applicability. Finally, we highlight the current limitations of autoregressive models, such as computational complexity and mode collapse, and propose potential directions for future research. In summary, this survey makes several contributions:

- Given the recent surge of advances based on visual autoregressive models, we provide a comprehensive and timely literature review of these models, aiming to offer readers a quick understanding of the generic autoregressive modeling framework.

- We categorize visual autoregressive models based on their sequence representation strategies and systematically compile applications across various domains. This aims to help researchers in specific fields quickly identify and learn about related work.

- We provide a comprehensive review of autoregressive models in vision from about 250 related references and summarize their evaluations compared with GAN/Diffusion/MAE-based methods in four image generation benchmarks (ImageNet, MS-COCO, MJHQ-30K, and GenEval bench).

## 2 Autoregressive Models

### 2.1 Preliminary

Visual autoregressive models are a class of generative models that sequentially predict visual elements, where each prediction is conditioned on the previously generated elements. Visual autoregressive models generally consist of two core components:

**Sequence Representation.** The visual data is first transformed into a sequence of discrete elements. These elements may correspond to pixels, image patches, or latent codes derived from visual content, depending on the specific model architecture. This transformation allows the visual data to be framed as a sequential modeling problem, akin to text generation in natural language processing.

**Autoregressive Sequence Modeling.** Once the visual content is represented as an ordered sequence, the model is trained to generate each element by conditioning all preceding elements. Mathematically, this is expressed as:

$$p(x) = \prod_{i=1}^{N} p(x_i | x_1, x_2, ..., x_{i-1}; \theta),\tag{1}$$

where $p_{(x_i|x_1, x_2, ..., x_{i-1}, \theta)}$ represents the probability of the current element $x_i$ conditioned on all previous elements in the sequence, with $\theta$ denoting the model parameters. The training objective is to minimize the negative log-likelihood(NLL) loss, which is formulated as:

$$\mathcal{L}(\theta) = -\sum_{i=1}^{N} \log p(x_i | x_1, x_2, ..., x_{i-1}; \theta).\tag{2}$$

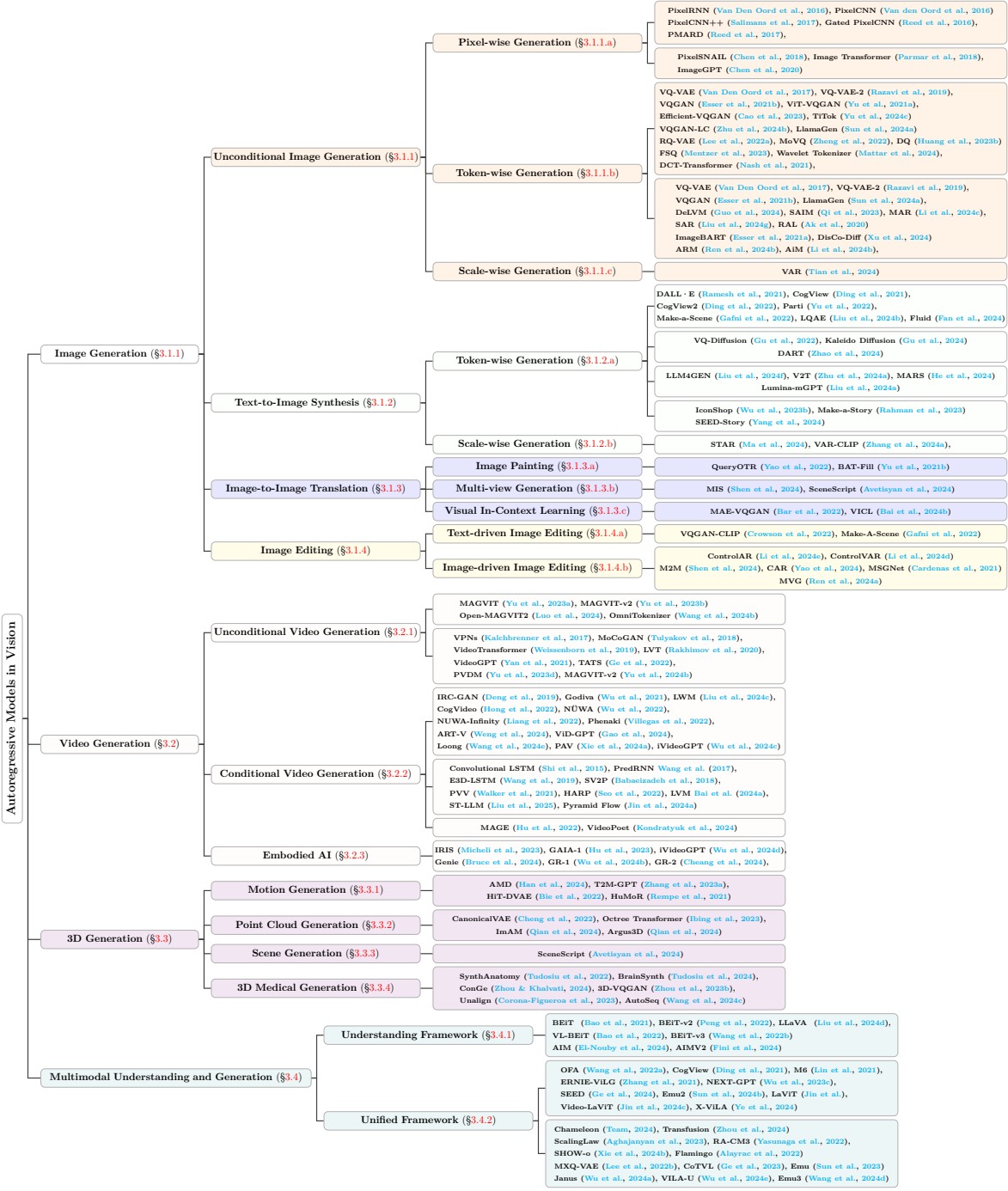

Figure 2: Literature taxonomy of autoregressive models in vision.

## 2.2 Generic Frameworks

In the preceding sections, we have provided a general overview of visual autoregressive models about the essential concept of sequence representation and sequence modeling. Then, we consider the representation strategy to classify the visual autoregressive models further, *i.e.* pixel-based models, token-based

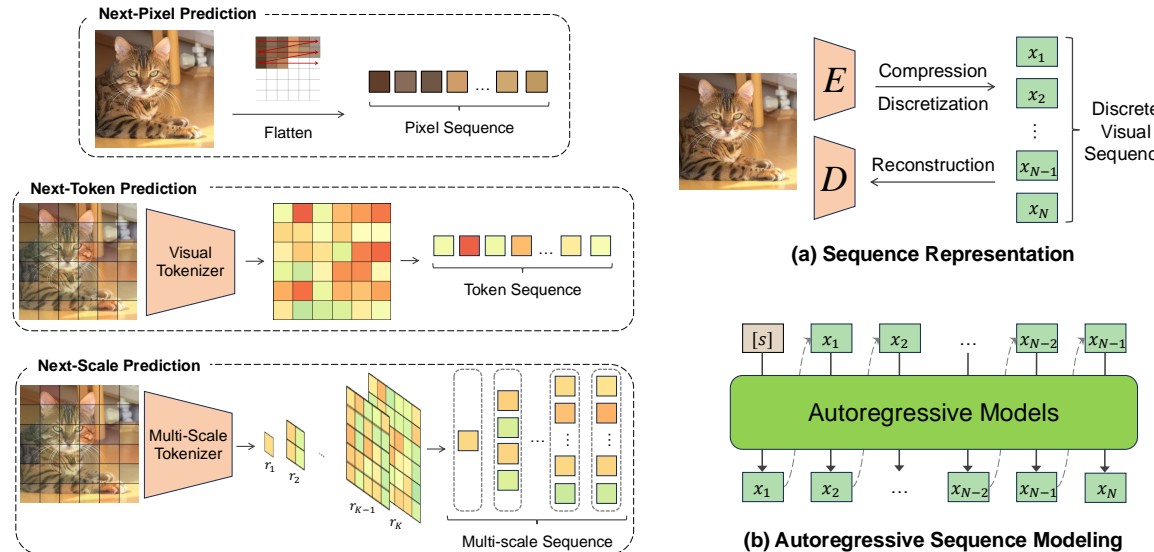

Figure 3: Illustration of three types of visual autoregressive models general frameworks based on their sequence representation strategies. **Next-Pixel Prediction** flattens the image into a *pixel sequence*. **Next-Token Prediction** converts the image into a *token sequence* via a visual tokenizer. **Next-Scale Prediction** employs a multi-scale tokenizer to generate a *multi-scale sequence*.

Figure 4: Core components in visual autoregressive models. (a) **Sequence Representation** encodes visual data into the discrete visual sequence, followed by reconstruction. (b) **Autoregressive Sequence Modeling** predicts each element sequentially.

models, and scale-based models. Since representation strategy directly reflects how models handle the underlying structure and features of visual data. Different representation strategies influence the model's granularity, efficiency, and applicability. Pixel-based models (Sec. 2.2.1) generate images pixel by pixel, capturing fine-grained spatial details. Token-based models (Sec. 2.2.2) represent images as a set of high-level discrete visual tokens, analogous to words in NLP. Scale-based models (Sec. 2.2.3) generate images through a multi-scale representation, allowing the model to handle visual content from different resolutions. We detail these three types of generic frameworks below.

**Discussion about continuous methods**  Our categorization of autoregressive generation methods is based on the representation strategy of the autoregressive elements. Although our discussion primarily focuses on discrete autoregressive methods, we note that continuous autoregressive approaches (Li et al., 2024c; Zhou et al., 2024), which operate directly in the embedding space without discretization, can be naturally accommodated within this framework. Continuous autoregressive models share many similarities with their discrete counterparts, with two notable differences: (a) they perform regression directly on continuous embeddings, eliminating the need for quantization; and (b) while discrete models typically minimize negative log-likelihood using a cross-entropy loss, continuous models require alternative loss formulations (e.g., diffusion loss (Li et al., 2024c)) on a case-by-case basis. Despite these differences, the underlying principles of autoregressive generation paradigms remain analogous, and our taxonomy is flexible enough to encompass both discrete and continuous methods.

### 2.2.1 Pixel-based Models

In pixel-based models, visual data is directly represented at the pixel level. PixelRNN (Van Den Oord et al., 2016) emerged as the pioneering work for AR image generation. It converts a 2D image $I \in \mathbb{R}^{H \times W \times 3}$ into a 1D discrete pixel sequence $\{x_1, x_2, \cdots, x_N\}$ by applying a raster scan order.

PixelRNN employs LSTM layers to sequentially predict each pixel based on previously generated pixels, as illustrated in Equation 1. Following this development, a series of AR generation works centered around pixelCNN (Van den Oord et al., 2016; Reed et al., 2016; Salimans et al., 2017; Chen et al., 2018; Reed et al., 2017) were proposed, achieving remarkable generation quality on CIFAR-10 and ImageNet $32 \times 32$. Building upon these advancements, several attempts (Kalchbrenner et al., 2017; Weissenborn et al., 2019) are made to extend the pixel-level AR approach to video generation through per-pixel synthesis.

However, despite these milestones, generating high-resolution images remains a significant challenge for pixel-level autoregressive models. This is primarily attributed to 1) the quadratically increasing computational cost with sequence length and 2) the redundancy inherent in per-pixel information. Some research (Reed et al., 2017) has explored the adoption of parallel techniques to generate images at $256 \times 256$ resolution. However, these methods tend to yield suboptimal and blurry results.

### 2.2.2 Token-based Models

The token-based models represent a significant evolution in visual autoregressive modeling, drawing inspiration from natural language processing(NLP). Unlike pixel-level models, which operate directly on raw visual data, this paradigm compresses and quantifies an image or video into a sequence of discrete tokens, allowing for more efficient processing of high-resolution content.

To achieve this, Vector Quantization(VQ) technique is employed to transform continuous visual features into a sequence of discrete latent codes. Given a raw image $x \in \mathbb{R}^{H \times W \times 3}$, an encoder $E$ first maps the image to a latent feature map $E(x) \in \mathbb{R}^{h \times w \times d}$. These continuous features are then quantized into discrete codes using a learned codebook $\mathcal{Z} = \{z_k\}_{k=1}^{K} \subset \mathbb{R}^d$ containing $K$ entries, where each entry $z_k$ represents a prototype vector in the latent space. The quantization operation is defined as finding the nearest codebook entry for each spatial feature vector $\hat{z}_{ij} \in \mathbb{R}^d$:

$$z_q(x) = \left( \arg \min_{z_k \in \mathcal{Z}} \|\hat{z}_{ij} - z_k\| \right) \in \mathbb{R}^{h \times w \times d}. \tag{3}$$

This quantization process enables a compact and discrete representation of the latent feature space, which is particularly advantageous for reducing the computational burden for high-dimensional image generation tasks. A seminal work on discrete representation is VQVAE (Van Den Oord et al., 2017), which introduced a two-stage paradigm that has become the *de-facto* standard for autoregressive vision generation. In this paradigm, an encoder-decoder architecture is initially trained to learn the discrete image representation. The encoder $E$ maps the input image $x$ to a latent space $z_e(x)$, where the continuous latent features are quantized into discrete codes using VQ. The decoder $D$ then reconstructs the image from these discrete codes. The training objective consists of a reconstruction loss and a commitment loss to ensure the latent codes effectively represent the input:

$$\mathcal{L} = \|x - D(z_q(x))\|_2^2 + \|sg[E(x)] - z_q(x)\|_2^2 + \beta \|E(x) - sg[z_q(x)]\|_2^2, \tag{4}$$

where $sg$ denotes the stop-gradient operator, $E(x)$ is the encoded latent vector, and $z_q(x)$ is the quantized vector.

In the second stage, a powerful autoregressive model is trained to predict the next discrete token given the sequence of previously generated tokens. Building upon the VQVAE, VQ-VAE-2 (Razavi et al., 2019) introduced a multi-scale hierarchical architecture that further enhances the quality and diversity of generated images. By incorporating multiple levels of latent representations, VQ-VAE-2 captures both global and local details. This hierarchical approach improves the model's ability to generate high-resolution images, achieving state-of-the-art results on various image datasets. VQGAN (Esser et al., 2021b) enhances the image tokenizer by integrating an additional PatchGAN-based discriminator loss into the VQVAE framework. This approach facilitates the learning of a perceptually rich codebook.

Furthermore, VQGAN employs a GPT-2-style decoder-only Transformer (Vaswani, 2017) to model the distribution of tokens in a raster-scan order, scaling autoregressive image generation with millions of pixels while maintaining computational efficiency.

### 2.2.3  Scale-based Models

The scale-based models, as proposed in VAR (Tian et al., 2024), introduce a hierarchical method for autoregressive image generation. Unlike traditional next-token prediction models that operate on a single resolution in a raster-scan order, it generates visual content across multiple scales from coarse to fine. The autoregressive unit in this approach is an entire token map instead of a single token, which enables the model to process visual data in a more structured and efficient manner.

A foundational idea behind VAR is the Residual Quantization(RQ) technique introduced in RQ-VAE (Lee et al., 2022a). RQ-VAE improves upon standard VQ by recursively quantizing residuals of feature maps. Unlike VQ-VAE, which requires larger codebooks to maintain quality as the resolution of the quantized feature map decreases, RQ-VAE uses a fixed-size codebook and quantizes a vector $z$ by approximating the residuals step-by-step in a coarse-to-fine manner:

$$\text{RQ}(z; C, D) = (k_1, k_2, \ldots, k_D), \quad \text{where } k_d = \arg\min_{z_i \in C} |r_{d-1} - z_i| \tag{5}$$

where $C$ is the shared codebook, $D$ is the quantization depth, and $r_{d-1}$ is the residual vector at depth $d-1$. This recursive quantization process allows RQ-VAE to represent high-resolution images compactly, reducing spatial resolution while retaining essential information.

Building upon this idea, VAR employs a multi-scale quantization autoencoder to discretize an image into token maps at various scales. Given a raw image $x \in \mathbb{R}^{H \times W \times 3}$, a multi-scale VQ-VAE encodes the image into a set of token maps $\{\boldsymbol{R}_1, \boldsymbol{R}_2, ..., \boldsymbol{R}_k\}$, where each token map $\boldsymbol{R}_k \in \mathbb{R}^{h_k \times w_k \times d}$ corresponds to a different scale $k$ of the image and serves as the autoregressive unit. The generative process is hierarchical, starting from the coarsest scale and autoregressively generating higher-resolution token maps. During the $k$-th autoregressive step, all distributions over the $h_k \times w_k$ tokens in $r_k$ will be generated in parallel.

Compared to token-based models, the scale-based models offer several advantages: a). It retains spatial locality, which facilitates zero-shot generalization to novel tasks without the need for task-specific training; b). The efficiency of token generation is improved by enabling parallel token generation within each token map, reducing the overall computational complexity. This efficiency gain is particularly beneficial for scaling up to larger image resolutions.

### 2.2.4  Analysis of Computational Costs

We compare the time complexity and efficiency of three autoregressive generation paradigms: next-pixel prediction, next-token prediction, and next-scale prediction. We consider a baseline "vanilla" architecture comprising a standard self-attention Transformer for autoregressive modeling, and an optional CNN-based tokenizer for decoding. Our analysis assumes greedy sampling without any additional efficiency techniques.

In next-pixel prediction, an $N \times N$ image is generated sequentially, with the Transformer's self-attention incurring a per-step cost of $O_T(i^2)$ for a sequence of length I. Consequently, generating all $N^2$ pixels requires:

$$\sum_{i=1}^{N^2} i^2 = \frac{1}{6} N^2(N^2 + 1)(2N^2 + 1) \tag{6}$$

which is equivalent to $O_T(n^6)$ basic computation. Next-token prediction reduces this cost by incorporating an image tokenizer with a compression ratio $k$, which shortens the latent sequence to $(N/k)^2$

|  | Require Tokenizer | Compression Ratio | Complexity | Efficiency |
|---|:---:|:---:|:---:|:---:|
| Next-Pixel Prediction | ✘ | - | $O_T(N^6)$ | ☆ |
| Next-Token Prediction | ✔ | $k$ | $O_T(N^6/k^6) + O_C(N^2)$ | ☆ ☆ |
| Next-Scale Prediction | ✔ | $k$ | $O_T(N^4/k^4) + O_C(N^2)$ | ☆ ☆ ☆ |

Table 1: Efficiency Comparison of Different Autoregressive Generation Paradigms. We consider the task of generating an $N \times N$ image using a standard self-attention Transformer and an optional CNN-based tokenizer.

tokens. The autoregressive cost is thereby reduced to $O_T(N^6/k^6)$, and decoding the latent representation back to an $N \times N$ image via a CNN incurs a cost of $O_c(N^2)$. The total cost is significantly lower than that of next-pixel prediction, particularly for high-resolution image synthesis. Next-scale prediction further enhances efficiency through a block-wise causal masking strategy and a reduced number of iterative steps, resulting in a time complexity of $O_T(N^4/k^4) + O_c(N^2)$ (Tian et al., 2024). Table 1 summarizes comparative computational complexities and efficiency ratings of these paradigms.

## 2.3 Relation to Other Generative Models

**Variational Autoencoder.** Variational Autoencoders (VAEs) (Kingma, 2013) are a class of generative models that learn to map data into a continuous, lower-dimensional latent space and subsequently reconstruct it back to the original data space. This process is governed by optimizing a variational lower bound of the data likelihood. In contrast, autoregressive models directly capture the full data distribution by predicting each element sequentially, optimizing negative log-likelihood(NLL) as the training objective. While VAEs provide efficient sampling, their reliance on variational approximations often results in less sharp outputs (Child, 2021) and may suffer from posterior collapse (Zhao et al., 2019; Chen et al., 2017) when equipped with a powerful decoder. Autoregressive models, on the other hand, generate high-quality samples by modeling data dependencies at the original dimensionality, but they are slower in inference due to their sequential generation process. To address these limitations, hybrid approaches (Gulrajani et al., 2017; Van Den Oord et al., 2017; Chen et al., 2017) have been proposed that integrate the autoregressive process into VAEs. One prominent example is VQ-VAE (Van Den Oord et al., 2017), which utilizes a VAE framework to learn discrete latent spaces and autoregressive models to refine generation, effectively leveraging the strengths of both models for efficient and high-quality image synthesis.

**Generative Adversarial Networks.** GANs (Goodfellow et al., 2014) are known for generating high-quality images with fast inference, enabled by their one-shot generation process. GANs demonstrate particular strength in domains with structured data, such as facial synthesis, where their low-dimensional latent space can effectively manipulate visual attributes (Xia et al., 2022). However, the adversarial training objective that GANs rely on often results in training instability and mode collapse (Liu et al., 2020), which requires a delicate balance between the generator and discriminator. Additionally, the adversarial training paradigm that underpins GANs can hinder their scalability to more diverse datasets and larger model sizes. In contrast, autoregressive models employ likelihood-based training, which ensures a stable training process. Despite their relatively slow sampling speeds, autoregressive models exhibit favorable scaling laws, where model performance consistently improves with larger datasets and increased model sizes (Kaplan et al., 2020). These properties make autoregressive models particularly well-suited for building universal generative models and can be adapted to a wide range of downstream tasks through zero-shot generalization of supervised fine-tuning (SFT), enabling flexibility and robustness across different applications.

**Normalizing Flows.** Normalizing flows (Dinh et al., 2014; Rezende & Mohamed, 2015) utilize a series of invertible and differentiable transformations to map a simple Gaussian distribution into a

complex data distribution. Each transformation is designed to have an easily computable Jacobian determinant, enabling efficient computation of the model's likelihood. Both normalizing flows and autoregressive models allow direct optimization via maximum likelihood estimation. However, normalizing flows achieves this by enforcing a tractable likelihood, which imposes specific architectural constraints to ensure invertibility. In contrast, autoregressive models optimize the likelihood by discretizing the data and sequentially predicting each element, thereby allowing greater flexibility in model design and enhanced scalability with data and model size.

**Diffusion Models.** Diffusion models (Ho et al., 2020) have emerged as the state-of-the-art generation paradigm for a wide range of vision generation tasks. As recent challengers, autoregressive models share several characteristics with diffusion models. Both methods can generate diverse, high-quality samples, yet both also suffer from inefficient inference due to their iterative or sequential generation processes. Additionally, both approaches are likelihood-based and optimize objectives such as Negative Log-Likelihood (NLL) or Evidence Lower Bound (ELBO), making them relatively easy to train. Recent advances (Esser et al., 2021b; Rombach et al., 2022) in both fields have focused on compressing visual content into latent spaces to improve the efficiency of high-resolution generation. Despite these similarities, there are fundamental differences in their generative paradigms. Diffusion models employ a predefined forward process that gradually corrupts data and an iterative denoising process to recover samples. This iterative process retains spatial locality at each step, which benefits visual coherence. However, the necessity to corrupt the training data with Gaussian noise may potentially limit their scalability for understanding tasks. Additionally, while diffusion models operate primarily in continuous spaces, their discrete variants still lag behind in performance, posing challenges for multimodal applications like text generation. Autoregressive models, by contrast, inherently introduce unidirectional biases and discretization, which might not be ideal for visual tasks, partially explaining their lag behind diffusion models in recent benchmarks. However, autoregressive models offer flexibility in handling diverse modalities and combining generation with understanding. This adaptability aligns with the emerging trend of integrating autoregressive models with Large Language Models (LLMs) to create a unified framework for multimodal input-output tasks, bridging both generation and understanding capabilities. These distinctions and complementarities between the two approaches suggest potential avenues for future research. Recent efforts have aimed to combine the strengths of both approaches, such as integrating bidirectional contexts (Esser et al., 2021a; Zhou et al., 2024; Xie et al., 2024b), autoregressive models with continuous representations (Li et al., 2024c; Zhou et al., 2024), discrete diffusion models (Gu et al., 2022), and using autoregressive models to enhance the understanding capabilities of diffusion models (Gu et al., 2024).

**Masked Autoencoder.** Masked autoencoders (MAEs) (He et al., 2022) are designed to learn robust data representations by randomly masking portions of the input data and training the model to reconstruct the missing content. Both MAEs and autoregressive models share similarities as they compress images into discrete sequences of visual elements and model these sequences. In fact, MAEs and autoregressive models inherit two leading paradigms from natural language processing (NLP): MAEs adopt the BERT-style (Devlin et al., 2018) encoder-decoder structure, while autoregressive models follow the GPT-Style (Radford et al., 2019) decoder-only approach. Nevertheless, there are essential differences between them: 1) MAEs are trained by randomly masking visual tokens and reconstructing the missing part, while autoregressive models are trained to predict the next element in an ordered sequence; 2) MAEs utilize full attention, enabling each token to consider the entire surrounding contexts, whereas autoregressive models typically employ causal attention, restricting each token to focus only on previous ones; 3) During decoding, MAEs randomly generate multiple tokens in parallel, while autoregressive models generate tokens sequentially in a predefined order. These differences lead to distinct strengths: MAEs generally excel in representation learning and visual understanding tasks, and recent works like MaskGIT (Chang et al., 2022) and MAGE (Li et al., 2023c) also extend the Masked Image Modeling (MIM) approach to image generation. In contrast, autoregressive models are more suited for generating high-quality samples from scratch and demonstrate stronger scaling

laws as they are scaled up, a property that has also been validated in language generation. Recent research (Li et al., 2024c) has proposed viewing masked generative models as a generalized form of autoregressive models and has investigated ways to enhance autoregressive models with MAE tricks. However, given the essential differences between them and the comprehensive discussions on MAEs in prior surveys, our review in this paper is mainly focused on standard autoregressive models. In summary, both methods are based on sequence modeling and have their respective strengths. MAEs provide efficient parallel decoding and the bidirectional attention in MAEs is well-suited for visual data, and autoregressive models excel in scaling up and adapting to multimodal tasks. Rethinking the relationship between these two approaches could potentially advance autoregressive generation. One method that bridges these two algorithms is Show-o (Xie et al., 2024b), which incorporates MAE-like masking strategies and bidirectional attention within an autoregressive framework to enhance image generation capabilities.

## 2.4 Comparison between Autoregressive Methods and Non-Autoregressive Methods

In this subsection, we discuss the unique advantages of autoregressive models for vision in comparison to non-autoregressive models, such as the popular diffusion-based (Ho et al., 2020) and rectified-flow based models (Liu et al., 2022). We summarize these advantages in three key points:

**1. Scaling Laws**: The success of the next-token prediction paradigm in NLP (Achiam et al., 2023; Lyu et al., 2023; Dubey et al., 2024) can be largely attributed to well-established scaling laws (Henighan et al., 2020; Kaplan et al., 2020). Although the scaling properties of methods like diffusion and GANs remain relatively underexplored, autoregressive visual models offer the potential to transfer successful scaling experiences from NLP. This could enable efficient scaling in visual generation frameworks, particularly in terms of model size and performance. Some work like VAR (Tian et al., 2024), Llama-Gen (Sun et al., 2024a) and FLUID (Fan et al., 2024) exemplify how autoregressive methods benefit from these scaling laws, highlighting their potential for advancing visual generation.

**2: Deployment Efficiency**: Autoregressive generative models can leverage the existing deployment technologies designed for language models (Wan et al., 2023). Frameworks such as VLLM provide acceleration for autoregressive models, significantly improving generation efficiency. This allows autoregressive visual models to benefit from infrastructure that has already been optimized for language tasks, facilitating seamless deployment in real-world applications.

**3. Bridging Language and Vision**: Autoregressive models offer a promising pathway toward unifying multimodal understanding and generation. By aligning closely with the structure of large language models, autoregressive approaches may provide a natural bridge between vision and language tasks. This synergy opens up the potential for more integrated and versatile models that can handle both vision and language processing in a unified manner, contributing to a deeper understanding of multimodal tasks (Team, 2024; Zhou et al., 2024; Xie et al., 2024b; Wu et al., 2024a).

In addition to these primary advantages, autoregressive models share some secondary benefits with certain non-autoregressive models. For example, autoregressive training demonstrates inherent stability. By directly optimizing the negative log-likelihood (NLL) through the minimization of the cross-entropy loss function, as depicted in Eq. 2, autoregressive models eliminate the need for adversarial training, as is the case with GANs (Goodfellow et al., 2014), or for the complex design of invertible networks like normalizing flows (Dinh et al., 2014; Rezende & Mohamed, 2015). Furthermore, autoregressive models often demonstrates well generation diversity, less worrying about the model collapse curse that can affect non-autoregressive models.

Despite these advantages, autoregressive models do have certain limitations. One significant drawback, when compared to diffusion models, is their difficulty in generating ultra-high-resolution images or videos. The higher-order time complexity of autoregressive generation makes this task more challenging. As discussed in Table 1, autoregressive models using next-token prediction paradigm have a

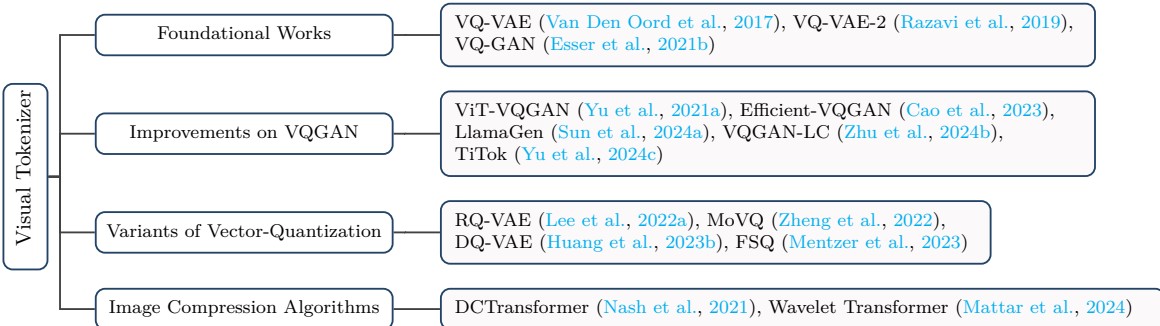

Figure 5: Taxonomy of visual tokenizer in an unconditional generation.

time complexity of $O(N^6/k^6)$ when generating an $N \times N$ image, whereas a Diffusion Transformer (DiT) model (Peebles & Xie, 2023) in a similar setting typically requires only $O(TN^4)$ time complexity (T for the denoising steps). Additionally, while autoregressive generation has shown promising progress in prompt-following when generating images from given prompts (Fan et al., 2024; Wang et al., 2024d), as demonstrated by benchmarks like GenEval (Ghosh et al., 2024), the visual quality of single images generated by autoregressive models still lags behind state-of-the-art diffusion or flow-based models (Esser et al., 2024; Labs, 2024), and significant research is still needed to bridge this gap. We provide a more detailed discussion of potential future directions for this work in Section 5.

## 3 Visual Autoregressive Models

### 3.1 Image Generation

#### 3.1.1 Unconditional Image Generation

Unconditional image generation refers to models producing images without specific input or guiding conditions, relying solely on learned visual patterns from the training data. This process can be categorized into three paradigms: pixel-level, token-level, and scale-level generation. Pixel-level generation creates images pixel by pixel, while token-level treats images as sequences of discrete tokens for more efficient generation. And scale-level generation builds images progressively from low to high resolution. This subsection will focus on the generation process itself, highlighting how models synthesize images by autoregressive generation of each part based on previously generated content, ensuring coherence and consistency. While other applications of autoregressive image generation will be introduced in subsequent sections.

#### 3.1.1.a Pixel-wise Generation

In the pixel-wise generation, the model constructs an image pixel on a pixel-by-pixel basis, unconditionally predicting each pixel's quantified value based solely on the sequence of previously generated pixels. This approach allows for direct optimization of the likelihood function, thus producing highly detailed outputs, albeit at a significant computational cost. This unconditional generation marks the inception of AR generation research, laying the foundational paradigm upon which all subsequent work in this domain has been built.

**Recurrent and Parallel Pixel Generation Techniques.** PixelRNN (Van Den Oord et al., 2016) uses RNNs to predict pixel values sequentially, effectively capturing complex dependencies between pixels. This work is the first to propose using an autoregressive approach for image generation. Building on its success, PixelCNN (Van den Oord et al., 2016) adopts dilated convolutions to efficiently capture

long-range pixel dependencies for generation. PixelCNN++ (Salimans et al., 2017) further optimizes PixelCNN using a discretized logistic mixture likelihood function and architectural improvements, improving image quality and efficiency. Reed et al. (2017) introduces a parallelized version of PixelCNN, which reduces computational complexity and allows faster high-resolution image generation.

**Transformer-based Approaches for Pixel Generation.** Traditional CNNs and RNNs face scaling issues when used for autoregressive image generation on large datasets, mainly due to their limited ability to model long-term dependencies, and their model architectures also struggle to scale up in size. To address this issue, Transformer (Vaswani, 2017) has been adapted for pixel-wise generation, leveraging its superior capacity to capture long-range dependencies within the data. PixelSNAIL (Chen et al., 2018) is the first work to combine causal convolutions with self-attention mechanisms, allowing the model to consider both local and broader contexts while preserving the sequential nature of pixel generation. Inspired by it, The Image Transformer (Parmar et al., 2018) employs self-attention to handle local neighborhoods in larger images, improving performance on complex tasks. ImageGPT (Chen et al., 2020) utilizes a decoder-only architecture and goes further by treating images as sequences of pixels, employing an autoregressive framework similar to that used in natural language processing. The above approaches demonstrate the power of Transformers in modeling long-range dependencies between pixels, laying groundwork for future advancements in large-scale unsupervised image learning.

### 3.1.1.b   Token-wise Generation

Token-wise generation utilizes a different approach, where the image is divided into a sequence of tokens, and each token is generated based on the previously generated tokens, similar to how text is processed in natural language models. This paradigm often leverages models like transformers, which can handle larger contexts in one step compared to pixel-by-pixel generation. These tokens typically represent local patches of the image in a more compact form, allowing for faster generation and scalability in terms of image size and complexity. Token-wise generation focuses on the representation of images as sequences of discrete tokens, i.e., the design of the image tokenizer, as well as the modeling of the autoregressive generation process at the token level.

### (1) Design of Image Tokenizer

**Brief Recap of Foundational Works.** In Sec 2.2.2, we have briefly introduced seminal works in visual tokenization, including VQ-VAE (Van Den Oord et al., 2017), VQ-VAE-2 (Razavi et al., 2019), and VQGAN (Esser et al., 2021b). These groundbreaking studies collectively established a two-stage paradigm for next-token prediction. This paradigm involves initially training a discrete visual tokenizer, followed by training a powerful sequence model to autoregressively predict the next token. VQ-VAE (Van Den Oord et al., 2017) initially introduces autoencoder-style training and VQ techniques, compressing a $128 \times 128$ image into a $32 \times 32$ discrete latent. Building on this, VQ-VAE-2 (Razavi et al., 2019) proposes a hierarchical structure that quantizes a $256 \times 256$ image into both $64 \times 64$ and $32 \times 32$ latent maps, corresponding to the bottom and top levels, respectively. VQGAN(Esser et al., 2021b) further advances this approach by integrating adversarial training and perceptual loss, achieving better perceptual quality and improving image compression rates. Due to its powerful representation capabilities, VQGAN is widely adopted in many autoregressive models and MLLMs, serving as a robust visual encoder.

**Improvements in Efficiency and Codebook Utilization.** Despite the advances brought by VQGAN, it still exhibits certain limitations, such as low codebook utilization and slow sampling speed. Several works have emerged to improve upon the vanilla VQGAN. Yu et al. (2021a) proposed ViT-VQGAN, which replaces the CNN network in the encoder-decoder architecture with a Vision Transformer (Dosovitskiy, 2020), demonstrating superior reconstruction quality. In addition, they introduce a linear projector to map latent codes from the high-dimensional space(768-d vector)

to a lower-dimensional space(32-d vector), facilitating efficient code index lookup. They also apply L2-normalization to the codebook vector to enhance training stability. ViT-VQGAN expands the codebook size from 1024 to 8196 without compromising the codebook utilization. Efficient-VQGAN (Cao et al., 2023) further improves efficiency by incorporating local attention through Swin Transformer blocks (Liu et al., 2021). TiTok (Yu et al., 2024c) propose a Transformer-based 1D tokenizer, built on ViT-VQGAN by introducing a set of fixed length latent tokens $K$ as input to the Vision Transformer. Adopting the architectures similar to Q-Former (Li et al., 2023b), this approach converts 2D image patches into 1D latent tokens of length $K$, enabling a more compact latent representation.

Zhu et al. (2024b) presents VQGAN-LC, which maintains a static codebook and trains a projector to map the entire codebook into the latent space without requiring modifications to the encoder/decoder. To initialize a static codebook, they utilize a pretrained CLIP image encoder (Radford et al., 2021) to extract patch-level features from the target dataset. These features are clustered, and class centers are selected to initialize the codebook. VQGAN-LC successfully scaled the codebook size of VQGAN to 100000 with a utilization rate of 99%. LlamaGen (Sun et al., 2024a) conducts a detailed ablation study on the codebook size and vector dimension in VQGAN. Their results reveal that the large codebook dimension and small size used in vanilla VQGAN resulted in each vector containing excessive information. Consequently, attempts to scale the codebook size resulted in poor utilization rates. Therefore, LlamaGen employs a much larger codebook size with smaller vector dimensions, which has been shown to significantly enhance both reconstruction quality and codebook utilization.

**Variants of Vector Quantization.** Beyond improvements on VQGAN, several works have also proposed variants of Vector Quantization. One such approach is Residual Quantization (RQ), introduced by Lee et al. (2022a). As previously discussed in Sec. 2.2.3 and Eq. 5, RQ represents an image as $D$ quantizing residuals of feature maps, recursively approximating the final latent code $z$ in a coarse-to-fine manner. Each depth $d$ shares the same codebook. Under constrained codebook sizes, RQ has been shown to outperform VQ in reconstruction performance and generation quality. Similarly, MoVQ (Zheng et al., 2022) adopts a multichannel index map with a shared codebook for image quantization. They introduce modulation operations during decoding to incorporate spatial variants and enhance reconstruction performance. Huang et al. (2023b) further propose Dynamic Quantization (DQ), employing a hierarchical encoder to represent images at multi-level. DQ employ Dynamic Grained Coding (DGC) modules to assign a dynamic granularity to each region, resulting in a multi-grained representation where each region has a variable length. Other works attempt to simplify the nearest-neighbor lookup in VQ by modifying the quantization process. Mentzer et al. (2023) propose Finite Scalar Quantization (FSQ), which assumes that the distribution of the latent code in a lower-dimensional space follows a simple, fixed grid partition. FSQ bounds each dimension of the latent code $z$ to $L$ values, avoiding complex nearest-neighbor computations. It prevents codebook collapse and demonstrates high codebook utilization when scaling up the codebook size.

**Inspiration from Image Compression.** Some works draw inspiration from traditional image compression algorithms to design more effective image tokenizers. DCT-Transformer (Nash et al., 2021), inspired by the JPEG compression algorithm (Wallace, 1991), designs a Discrete Cosine Transform(DCT) (Ahmed et al., 1974) based transformer to convert images into quantized DCT coefficients. Similarly, Mattar et al. (2024) develops a wavelet-based image coding method leveraging wavelet Tokenizer (Daubechies, 1992). This approach tokenizes visual details in a coarse-to-fine manner, ordering the information starting with the most significant wavelet coefficients. These approaches provide fresh perspectives on the design of image tokenizers.

## (2) Autoregressive Modeling

**Foundation Work in Autoregressive Modeling.** As previously discussed, VQ-VAE (Van Den Oord et al., 2017) and VQ-VAE-2 (Razavi et al., 2019) introduced the concept of tokenization, steering autoregressive visual generation towards a two-stage paradigm. Despite these advancement,

these models still rely on CNNs for token-wise autoregressive modeling. The inherent limitation of CNNs, due to their local attention mechanisms, impedes their ability to effectively model long sequence dependencies. A significant milestone that unlocked the potential of autoregressive visual generation was the advent of VQ-GAN (Esser et al., 2021b), which combines Transformer(Vaswani, 2017) with next-token prediction, employing a GPT-2 style decoder-only Transformer to predict subsequent tokens in a raster-scan order. VQGAN enables the scaling of autoregressive image generation to high-resolution images comprising millions of pixels, and surpasses all previous methods in both generation quality and resolution. Since VQ-GAN, Transformer have become the de-facto standard for autoregressive modeling.

**Scaling Up to Larger Models.** Following the establishment of Transformer structures, researchers have sought to investigate whether the scaling laws observed in language models are also applicable to autoregressive vision generation. LLamaGen (Sun et al., 2024a), for instance, directly employs the Llama architecture (Touvron et al., 2023a;b) without additional modifications and trains a series of image generation models with parameters ranging from 111M to 3.1B. Results demonstrate performance improvements with the increase in model parameters, indicating the potential for scalable image generation. In contrast, DeLVM (Guo et al., 2024) concentrated on minimizing data and parameter requirements to develop data-efficient large vision models. DeLVM achieves high adaptability across diverse vision tasks with significantly reduced data.

**Efficiency and Long-sequence Modeling.** Linear attentions have recently gained popularity for their efficiency in long-sequence modeling during autoregressive generation. Both ARM (Ren et al., 2024b) and AiM (Li et al., 2024b) leverage the Mamba architecture to enhance image generation. ARM (Ren et al., 2024b) has made successful exploration on combining autoregressive models with the Mamba architecture to enhance the visual capabilities of Mamba and accelerate the training process. AiM (Li et al., 2024b) applies Mamba directly for next-token prediction in autoregressive models, achieving superior quality and faster inference on ImageNet1K with a FID of 2.21x.

**Exploration Beyond Raster Order.** Traditional autoregressive methods typically serialize 2D images into 1D discrete sequences following a raster order. However, recent research suggests that this reliance on raster order and discrete representations may not be essential for autoregressive generation modeling. Studies indicate that exploring alternative methods could lead to improved performance and open up new possibilities for image representation. SAIM (Qi et al., 2023) introduces a stochastic process for autoregressive image modeling by predicting image patches in random orders rather than following a fixed raster order of pixel predictions. This stochastic approach challenges the conventional necessity of raster order in autoregressive modeling. MAR (Li et al., 2024c) provides a theoretical analysis suggesting that raster order and discrete representations are not necessary. MAR further introduces a diffusion loss and trains an autoregressive model(in general definition) with continuous representations in a Masked Image Modeling (MIM) style. They achieve comparable performance to traditional discrete autoregressive techniques, questioning the conventional reliance on discrete tokenization. SAR (Liu et al., 2024g) further generalizes the problem into a unified framework. SAR enables causal learning with any sequence order or output intervals, offering a more flexible and potentially powerful approach to autoregressive image modeling.

**Intergration with Other Generative Models.** The idea of combining autoregressive models with other generative models like GANs (Goodfellow et al., 2014) and Diffusion (Ho et al., 2020) has also garnered researchers' interest. RAL (Ak et al., 2020) innovatively introduces adversarial learning and policy gradient optimization into the training of autoregressive models to address the inherent exposure bias problem, which is a common issue in autoregressive models. ImageBART (Esser et al., 2021a) employs a coarse-to-fine autoregressive method, combining multinomial diffusion with hierarchical structures to iteratively improve image fidelity and details during generation. DisCo-Diff (Xu et al., 2024) aims to enhance diffusion models by augmenting them with learnable discrete latents. They

adopt an autoregressive transformer to model the distribution of these discrete latents. This approach achieves state-of-the-art FID scores on the ImageNet benchmark.

### 3.1.1.c   Scale-wise Generation

While next-token prediction has achieved considerable success in vision generation and has almost become the de-facto standard paradigm for visual autoregressive models, several significant challenges still warrant discussion. The first is computational efficiency. Due to the inherent 2D nature of images, the quadratic computational complexity of self-attention in transformers, and the autoregressive steps involved in generation. the overall complexity can reach $O(n^6)$ when generating an $n \times n$ token map. Although efforts have been made to reduce the sequence length through tokenization, the growing image resolution exacerbates this issue, rendering the computational demands increasingly unsustainable. The second challenge is the flattening operation. In the context of next-token prediction, visual tokens are flattened into a 1D sequence after tokenization, which limits the model's ability to leverage spatial locality during autoregressive generation. While recent studies (Sun et al., 2024a) demonstrate that LLM-style autoregressive models can produce high-quality images, the necessity of preserving spatial locality remains an open issue.

In contrast to next-token prediction, the next-scale prediction paradigm utilizes token maps at different scales as the base autoregressive units, progressively generating visual content via a coarse-to-fine manner. This paradigm further reduces the sequence length required for autoregressive generation. Generating a complete token map at each step, enables leveraging the inductive biases inherent in images to enhance the quality of visual generation. In line with previous sections, we introduce the design of visual tokenizers and autoregressive modeling within the next-scale prediction paradigm.

### (1) Design of Video Tokenizer

The paradigm of next-scale prediction, akin to next-token prediction, requires a visual tokenizer to compress an image into a discrete token map. A foundational technique behind this is the Residual Quantization (RQ) introduced in RQ-VAE (Lee et al., 2022a). RQ-VAE is initially developed to address the low utilization issue of VQGAN when scaling to larger codebook sizes, which is achieved by introducing an additional depth dimension $d$. In RQ-VAE, each latent vector is quantized into $D$ codes, with each code representing the quantization of the residual vector at the current depth. All depths $d$ share the same codebook. RQ recursively approximates the final latent vector $z$ in a coarse-to-fine manner, providing a more accurate approximation of $z$ while maintaining the codebook size. Note that RQ still quantizes the image into a token-wise sequence based on spatial positions, with each position having multiple residual codes at different scales. Although RQ is not a next-scale prediction method, its hierarchical approach offers valuable insights for next-scale prediction.

Building on the hierarchical and residual-style quantization concepts of RQ-VAE, VAR (Tian et al., 2024) introduces a more concise scale-wise quantization method. VAR employs a vanilla VQ-GAN (Esser et al., 2021b) to encode the feature map in the latent space. It then interpolates and computes residuals to quantize the feature map $f \in \mathbb{R}^{h \times w \times c}$ into $K$ multi-scale token maps $(r_1, r_2, ..., r_K)$, each at increasingly higher resolutions $h_k \times w_k$, culminating in $r_K$ which matches the original feature map's resolution $h \times w$. Similar to RQ-VAE, VAR quantizes the residuals of each scale relative to the previous scale, with residuals sharing the same codebook across different scales. Through this hierarchical approach, VAR represents an image as a coarse-to-fine scale-wise sequence. The quantization process of this sequence is entirely causal, and each autoregressive unit maintains spatial locality.

### (2) Autoregressive Modeling

Scale-wise generation involves creating images at multiple scales or resolutions. The model typically starts by generating a coarse, low-resolution version of the image and progressively enhances details through successive stages. This technique allows the model to capture both global structure and finer details without needing pixel-level precision in the initial stages. As each phase refines the image, it adds more detail and adjusts local features to improve overall fidelity and texture. Scale-wise generation is particularly effective for high-resolution image synthesis, where maintaining consistency across scales is crucial.

Despite its advantages, scale-wise generation is less common in autoregressive models due to the significant complexity it introduces in training and computation. Generating images at multiple scales requires careful coordination between different resolutions, which can be challenging and resource-intensive. Each scale builds upon the previous one, increasing the risk of error propagation from lower resolutions. Furthermore, this approach demands more computational power and memory as the model processes and refines images on several scales, making it less efficient than single-resolution methods.

A pioneering work that achieved this breakthrough is VAR (Tian et al., 2024), which brought the concept of "next-scale prediction" to reality. VAR innovatively shifts the focus from traditional pixel-wise prediction to a scale-wise generation paradigm. As previously introduced, VAR draws inspiration from RQ-VAE, quantizing a single image into token maps of various resolutions from coarse to fine. VAR employs a standard decoder-only transformer to model the next-scale prediction, while a block-wise causal mask is applied to enable each scale to depend only on its prefix and allow for parallel token generation. VAR outperforms existing autoregressive models and diffusion transformers, achieving superior results on benchmarks such as ImageNet. Furthermore, the model exhibits power-law scaling laws similar to those observed in large language models, which may reveal a new research direction for vision autoregressive modeling.

Overall, these scale-wise generation techniques mark significant advancements in high-resolution image synthesis. Following VAR, numerous works have emerged based on VAR including text-to-image generation, image editing, and controllable image generation. We will discuss these developments in the subsequent chapters.

### 3.1.2 Text-to-Image Synthesis

Text-to-Image generation involves generating images based on specific textual conditions provided to the model. Unlike unconditional image generation, which produces images without any input constraints, text-to-image generation leverages additional information to guide the creation process. By incorporating textual conditions, the model can produce more targeted and contextually relevant images, enhancing the quality and applicability of the generated outputs. The illustration of typical text-to-image synthesis frameworks is shown in Figure 6.

### 3.1.2.a Text-wise Generation

**Exploration across the integration of textual conditions and visual modeling.** A pioneering effort in this field is is exemplified by DALL · E (Ramesh et al., 2021), which employs a BPE-encoder and a pretrained VQ tokenizer to convert text and images into discrete tokens, respectively. The model then utilizes text tokens as a prefix condition and learns to predict the subsequent image tokens. Building upon this foundation, CogView (Ding et al., 2021) scales up the autoregressive token-wise model by employing a VQ-VAE tokenizer to transform image patches into discrete tokens. This approach enables the deployment of a 4-billion parameter Transformer model with higher performance. CogView2 (Ding et al., 2022) further explores a hierarchical strategy for text-to-image generation. It first train a base T2I model to generate images at low-resolution, followed by a fine-tuning stage aimed at super-resolution. This hierarchical approach successfully reduces computational demands and enhances large-scale image generation.

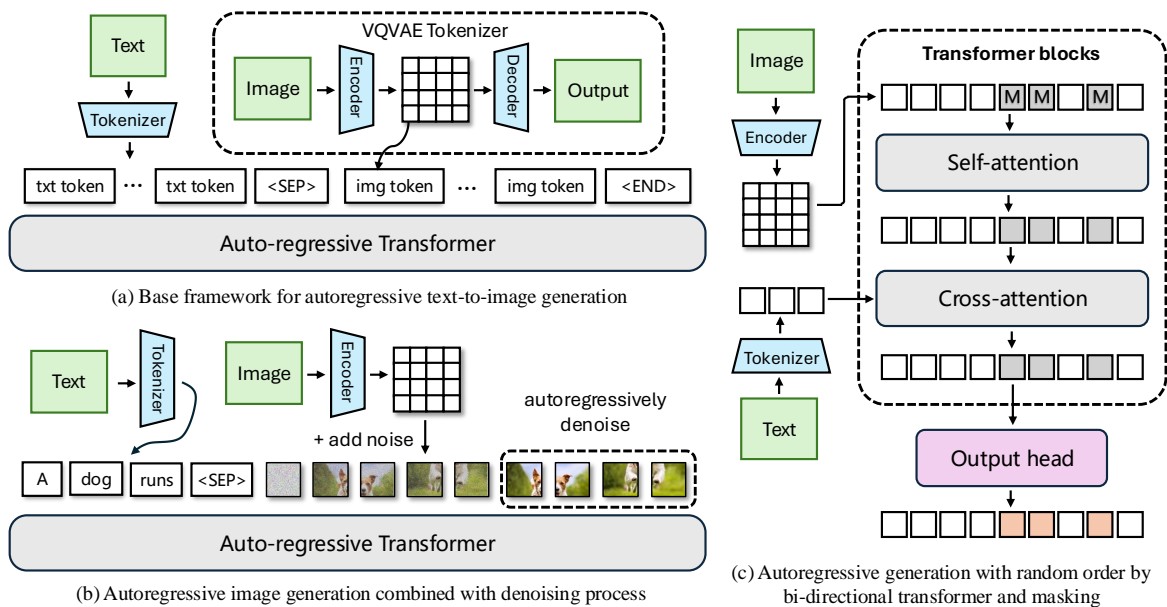

Figure 6: Illustration of **different frameworks for auto-regressive text-to-image generation**. (a) Generally, autoregressive text-to-image generation involves tokenizing text and images, and model all the tokens with a autoregressive transformer. (b) With the flexibility of autoregressive transformers, the autoregressive generation can be combined with other process, such as the denoising process or multi-scale generation process. (c) With the bi-directional transformer, the autoregressive generation sequence does not need to be a pre-defined order (such as raster scan), but can be any random order.

Actually, these methodologies treat text tokens as prefixes for autoregressive modeling, thereby the transformer is responsible for both capturing text semantics and generating image content within the autoregressive framework. In addition to this technical approach, another line of research involves encoding text tokens using a dedicated text encoder and generating image content using a transformer decoder. For instance, Make-a-Scene (Gafni et al., 2022) employs an external text encoder to process the textual input, while Parti (Yu et al., 2022) adopts a sequence-to-sequence (seq2seq) model with an encoder-decoder architecture. In Parti, a transformer encoder is utilized to encode the text information, and a transformer decoder is employed to generate the corresponding image. During a period when autoregressive modeling had not yet reached its full potential, these approaches demonstrated a more robust semantic understanding compared to earlier methods.

Furthermore, LQAE (Liu et al., 2024b) proposes an unsupervised technique for aligning text with images by quantizing image embeddings into text-like tokens, enabling few-shot multimodal learning. Fluid (Fan et al., 2024) introduces a random-order autoregressive model on continuous tokens, highlighting importance of token representation and generation order in scaling autoregressive models.

**Integration with Diffusion Models.** Inspired by the powerful generative capabilities of diffusion models, recent research has investigated the integration of diffusion models with autoregressive models for text-to-image generation. VQ-Diffusion (Gu et al., 2022) employs a Vector Quantized Variational Autoencoder (VQ-VAE) to map images into a discrete latent space. Subsequently, it utilizes a discrete diffusion decoder instead of a transformer decoder to reconstruct the images. In contrast, Kaleido Diffusion (Gu et al., 2024) maintains diffusion as the core generative framework while incorporating an autoregressive model to handle the latent conditions, thereby enhancing both text understanding and image generation capabilities. Another integrating autoregressive model with diffusion is DART (Zhao

et al., 2024), which unifies autoregressive and diffusion within a non-Markovian framework, achieving scalable, high-quality image synthesis.

**Integration with Large Language Models.** A significant advancement, LLM4GEN (Liu et al., 2024f), presented an end-to-end framework combining text encoders like CLIP with diffusion models, enriching token-wise generation with the semantic depth of large language models (LLMs). This results in improved semantic correspondence between text and image tokens for complex prompts. V2T (Zhu et al., 2024a) translates images into discrete tokens from an LLM's vocabulary, aligning visual and textual data for tasks such as image denoising. MARS (He et al., 2024) features the Semantic Vision-Language Integration Expert (SemVIE) module, deeply integrating textual and visual tokens, illustrating the flexibility of token-wise generation in producing detailed images from textual descriptions. Lumina-mGPT (Liu et al., 2024a) applies Flexible Progressive Supervised Finetuning (FP-SFT) to Chameleon (Team, 2024) with high-quality image-text pairs to fully unlock the potential of the model for high-aesthetic image synthesis while preserving its general multimodal capabilities.

**Expansion to Novel Tasks.** IconShop (Wu et al., 2023b) further extends token-wise generation to vector graphics, enabling scalable vector icon creation from text prompts. Make-a-story (Rahman et al., 2023) focuses on generating visually coherent stories from text, employing a scale-wise generation process to maintain consistency across multiple image frames within complex narratives. SEED-Story (Yang et al., 2024), a multimodal storytelling model, generates images and text in parallel, ensuring that visual details correspond seamlessly with the narrative across different resolution levels. Its hierarchical attention mechanism preserves coherence between text and images at all scales, from coarse descriptions to intricate image details.

### 3.1.2.b   Scale-wise Generation

As previously discussed, VAR pioneered a novel paradigm beyond token-wise generation, termed scale-wise generation, which generates images progressively from coarse to fine scales. Due to its advantages in efficient generation, coupled with its demonstration of scaling laws analogous to those observed in language models. Scale-wise generation has garnered significant research interest. Efforts have been made towards extending this paradigm to the text-to-image generation domain.

STAR (Ma et al., 2024) is a pioneering work in the domain of scale-wise text-to-image generation. Building upon the common paradigm of diffusion models (Rombach et al., 2022) in image generation, STAR employs features from a pretrained text encoder, utilizing cross-attention layers to provide detailed textual guidance. In addition, STAR introduces novel techniques such as Rotary Position Embedding (ROPE) (Su et al., 2024) to enhance the autoregressive modeling capabilities. These modifications result in improved text alignment and more efficient generation processes, demonstrating the potential of scale-wise text-to-image generation to achieve higher fidelity and coherence.

Concurrently, VAR-CLIP (Zhang et al., 2024a) introduces their approach by concatenating text embeddings from a pretrained CLIP model with visual embeddings from a VAR encoder. In this framework, the text embeddings serve as a condition to guide the generation of multi-scale tokens and the final image. This method has demonstrated notable effectiveness in text-to-image generation.

In summary, scale-wise generation models provide a powerful framework for generating high-resolution images by progressively refining outputs across multiple levels of detail. This hierarchical approach not only ensures global coherence, but also captures intricate details, making it a robust solution for text-to-image generation tasks.

### 3.1.3   Image-condition Synthesis

Autoregressive models have played a significant role in advancing image generation tasks, including Image-to-Image translation, where the goal is to translate one image domain to another.

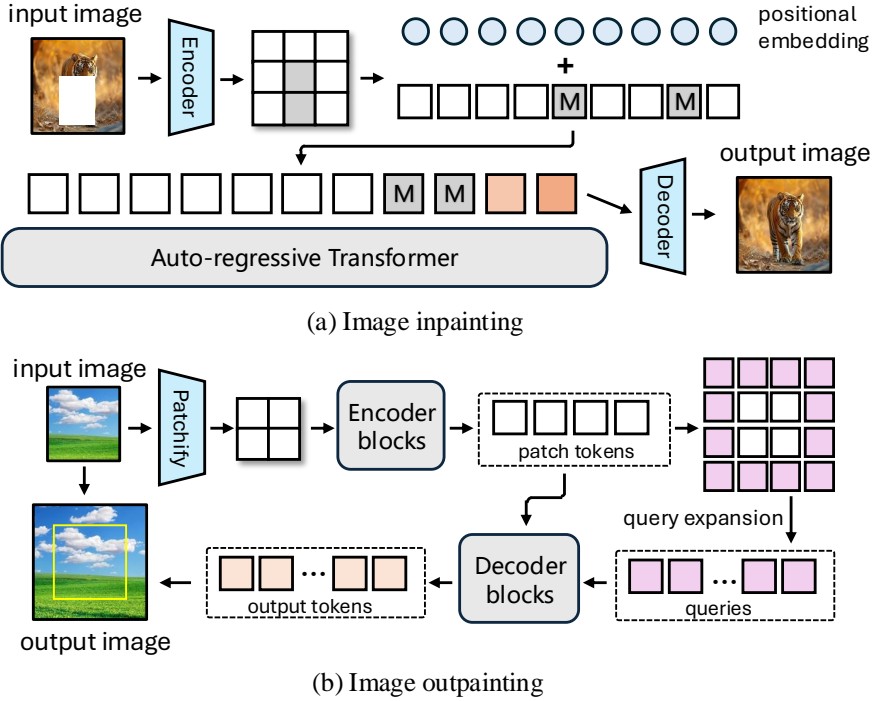

(a) Image inpainting

(b) Image outpainting

Figure 7: Frameworks for image painting. (a) **BAT-Fill** (Yu et al., 2021b) inpaints the image by rearranging the input tokens and auto-regressively generate output. (b) **Query-OTR** (Yao et al., 2022) does image outpainting by expanding input tokens into query tokens.

### 3.1.3.a Image Painting

In parallel, some works focuses on specific editing tasks such as inpainting and outpainting, where missing or extended visual content needs to be generated coherently. Inpainting fills in missing regions by seamlessly blending new content with the surrounding areas. Outpainting, on the other hand, extends image boundaries by generating new content that aligns naturally with the original scene, creating larger images from smaller inputs. Despite these differences, both techniques face challenge of maintaining visual coherence with existing context. Outpainting (Zhang et al., 2024b) is particularly useful for generating panoramic views or expanding image contexts. QueryOTR (Yao et al., 2022) addresses this by using a hybrid transformer-based encoder-decoder architecture, reframing outpainting as a patch-wise sequence-to-sequence task. For inpainting, Yu et al. (2021b) combines autoregressive modeling with context aggregation, ensuring consistent reconstruction of missing image regions by using the information surrounding the pixel. The overview of image painting are shown in Figure 7.

### 3.1.3.b Multi-view Generation

MIS (Shen et al., 2024) introduces an autoregressive framework for generating multiple interrelated images from the same distribution. By leveraging latent diffusion models, MIS enhances diversity while ensuring semantic coherence across scenes. This approach demonstrates the potential of autoregressive processes in multi-view and sequential image generation tasks. SceneScript (Avetisyan et al., 2024) further explores autoregressive modeling in the context of scene reconstruction, using token-based generation to translate complex 3D scenes into structured commands. Both works illustrate how autoregressive models can be extended beyond single-image generation to more complex tasks like multi-view generation and structured scene reconstruction.

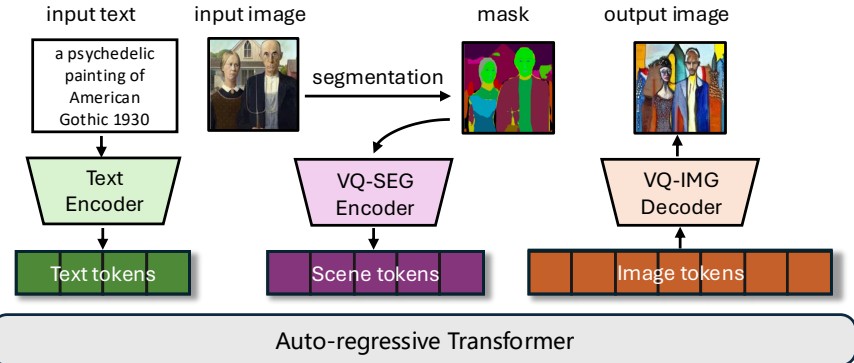

Figure 8: For autoregressive image editing, **make-a-scene** handles text tokens, scene tokens and image tokens with a autoregressive transformer.

### 3.1.3.c  Visual In-Context Learning

Bar et al. (2022); Bai et al. (2024b) has emerged as a key prompting method for visual autoregressive models. MAE-VQGAN (Bar et al., 2022) firstly proposes visual prompting, where a pretrained visual model adapts to new tasks as image inpainting without finetuning or modification. Inspired by it, VICL (Bai et al., 2024b) adapts in-context learning to vision tasks by presenting task descriptions or demonstrations through images or natural language text, encouraging models to generate predictions based on provided image queries.

### 3.1.4  Image Editing

Image editing is a crucial area in computer vision, focusing on modifying, enhancing, or reconstructing images based on various user inputs. Recent advancements in deep learning have enabled models to perform complex editing tasks such as inpainting, outpainting, and style transformations, allowing users to manipulate visual content with greater ease and precision. These tasks often involve generating new content that seamlessly integrates with the existing image, either by filling in missing parts (inpainting) or extending the image boundaries (outpainting), while maintaining visual consistency.

### 3.1.4.a  Text-driven Image Editing

VQGAN-CLIP (Crowson et al., 2022) introduces a method for generating and editing images using natural language input. Its editing capabilities stem from the combination of VQGAN's image synthesis and CLIP's ability to guide image modifications based on text prompts. This allows users to modify existing images or create new ones by altering styles, adding elements, or transforming parts of an image, while maintaining visual coherence. In contrast, Make-A-Scene (Gafni et al., 2022), illustrated in Figure 8, enhances this by incorporating scene layouts (segmentation maps) alongside text input. This addition enables more precise control over the structure and content of images, making it especially useful for localized edits. Make-A-Scene provides control over both semantics and spatial arrangement, whereas VQGAN-CLIP focuses more on creative, text-driven modifications.

### 3.1.4.b  Image-driven Image Editing

Image-driven image editing encompasses a range of techniques focused on modifying visual content while maintaining consistency with the original image's context and style. In the domain of autoregressive image generation, the introduction of control mechanisms is pivotal for enhancing both the flexibility and precision of the generated images. For example, ControlAR (Li et al., 2024e) integrates

spatial controls like canny edges and depth maps into the decoding process, allowing for a detailed and precise manipulation of image tokens. This approach enables more accurate alignment with the original image's features during generation. Similarly, ControlVAR (Li et al., 2024d) advances this concept by modeling image and pixel-level control representations through a scale-level strategy, thus achieving fine-grained control over the image synthesis process while preserving the inherent properties of autoregressive models. CAR (Yao et al., 2024) elaborates on a similar concept, focusing on advanced control mechanisms in autoregressive models to enhance the detail and adaptability of visual outputs.

On the other hand, for complex image generation tasks involving multiple objects, Many-to-Many Diffusion (M2M) (Shen et al., 2024) applies an autoregressive diffusion model to generate multi-image sequences, scaling across both spatial and temporal dimensions while maintaining consistency across frames. MSGNet (Cardenas et al., 2021) utilizes a vector-quantized variational autoencoder (VQ-VAE) combined with autoregressive modeling. This framework prioritizes both spatial and semantic coherence within generated images, effectively preserving consistency across multiple objects. Extending this concept to medical imaging, MVG (Ren et al., 2024a) proposes a unified framework for diverse 2D medical tasks such as cross-modal synthesis, image segmentation, denoising, and inpainting. This approach leverages autoregressive training conditioned on prompt image-label pairs, treating these tasks as image-to-image generation problems.

Overall, image-driven image editing techniques focus on refining image synthesis and manipulation tasks, employing control mechanisms, context-aware models, and coherent extensions of existing content to achieve precise and visually consistent outputs.

## 3.2 Video Generation

Building upon the advancements of autoregressive models in image generation, video generation (Yuan et al., 2024a;b) has similarly seen significant progress by extending these models to capture temporal dynamics in addition to spatial patterns. While autoregressive image models focus on generating static frames, video generation introduces the challenge of producing coherent sequences over time. Like image generation, video generation is categorized into unconditional and conditional approaches with different condition inputs. By leveraging the success of autoregressive models in both domains, researchers continue to push the boundaries of generating realistic and temporally consistent video sequences. Additionally, the integration of video generation techniques into embodied AI systems presents a new frontier, where generated videos are not just an end in themselves but are used to enhance the capabilities of intelligent agents in complex, real-world environments. The basic pipeline of autoregressive video generation can be found in Figure 9.

### 3.2.1 Unconditional Video Generation

**(1) Design of Video Tokenizer**

Recent works have expanded visual tokenization to videos by incorporating temporal compression. MAGVIT (Yu et al., 2023a) extends the 2D VQGAN into a 3D tokenizer to quantize video data into spatial-temporal visual tokens, employing an inflation technique for initialization using image pre-training. However, it struggles to tokenize images effectively. Building on MAGVIT, MAGVIT-v2 (Yu et al., 2023b) introduces a lookup-free quantization (LFQ) method, replacing the traditional codebook with binary latents derived from an integer set of size $K$, similar to word tokenizers in natural language processing (Sennrich, 2015). This LFQ approach allows the vocabulary size to grow in a way that benefits the generation quality of autoregressive models. Additionally, MAGVIT-v2 integrates C-ViViT (Villegas et al., 2022) and MAGVIT within a causal 3D CNN, enabling unified tokenization of both images and videos using a shared codebook. Open-MAGVIT2 (Luo et al., 2024) has successfully implemented and open-sourced MAGVIT-v2 for image tokenization, explored its application in auto-regressive models, and validated its scalability properties. OmniTokenizer(Wang et al., 2024b)

employs a transformer-based architecture with decoupled spatial and temporal blocks to perform image and video tokenization within a unified framework, utilizing a progressive training strategy to enable general-purpose visual encoding across various modalities.

## (2) Autoregressive Modeling

Unconditional video generation focuses on creating video sequences from scratch, without any specific input conditions. One of the foundational approaches in this area is the Video Pixel Networks (VPNs) (Kalchbrenner et al., 2017), which extend the PixelCNN model to video data. VPNs model each pixel in a video frame based on the preceding pixels and frames, allowing for the capture of intricate spatial details. However, VPNs face challenges in maintaining temporal coherence over longer sequences, resulting in less realistic motion when generating extended videos.

To address some of these limitations, MoCoGAN (Tulyakov et al., 2018) introduces an approach by decomposing the video generation process into two distinct components: motion and content. The model employs Generative Adversarial Networks (GANs) to separately generate a static content frame and a sequence of motion vectors, which are then combined to form the final video. This decomposition allows MoCoGAN to produce videos with more realistic and temporally coherent motion, although the strict separation of motion and content can sometimes limit the model's ability to generate highly complex and expressive content.

Video Transformer (Weissenborn et al., 2019) focuses on leveraging the transformer architecture to enhance the capacity and performance of these models to handle larger datasets and generate higher-resolution, longer videos. These advancements push the boundaries of what is possible with autoregressive models, enabling the generation of more detailed and extended video sequences. This approach offers a more efficient alternative to VPNs and MoCoGAN, particularly in handling the temporal dynamics of video generation.

LVT (Rakhimov et al., 2020) and VideoGPT (Yan et al., 2021) represent a significant step forward by combining the power of VQ-VAE with transformers. By operating in a discrete latent space, they achieve a balance between computational efficiency and high-quality video generation, improving both the diversity and fidelity of the generated videos.

To reflect the latest developments, including long-duration video synthesis, models such as TATS (Ge et al., 2022) have been introduced. This model leverages the combination of VQGAN and transformer architectures to produce temporally consistent long videos. Additionally, the PVDM (Yu et al., 2023d) demonstrates advanced diffusion-based approaches for video generation, offering improved probabilistic modeling in latent spaces. Furthermore, MAGVIT-v2 (Yu et al., 2024b) emphasizes the role of tokenizers in refining autoregressive models for video generation. LARP (Wang et al., 2024a) uses a learned AR generative prior, capturing global semantic content and enhancing compatibility with AR models.

### 3.2.2 Conditional Video Generation

Conditional video generation involves generating videos based on specific inputs, such as text descriptions, images, or existing video frames. This task is more complex than unconditional generation, as it requires the model to ensure that the generated content aligns with the given conditions.

**Text-to-Video Synthesis.** Text-to-video synthesis is challenging because it involves translating textual input into coherent and contextually appropriate video sequences. One of the earlier models addressing this challenge is IRC-GAN (Deng et al., 2019), which employs a recurrent architecture combined with GANs to iteratively refine video generation from text. By using introspective modules that encourage the model to correct its own mistakes and recurrent convolutional layers to handle temporal dependencies, IRC-GAN improves the quality and coherence of generated videos, setting a foundation

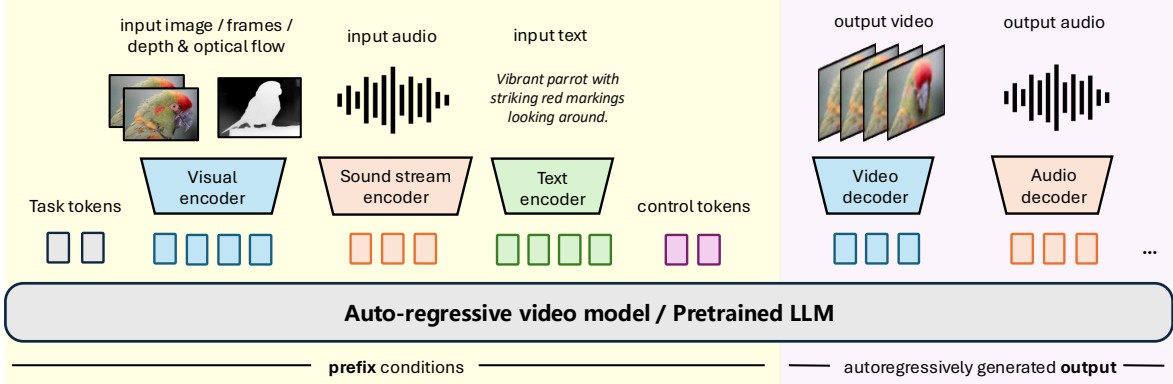

Figure 9: The pipeline of the auto-regressive video generation. The multi-modal input tokens are tokenized by different encoders, and set as prefix conditions for autoregressive models. The output tokens are autoregressively generated.

for future models. Godiva (Wu et al., 2021) utilizes transformers to align textual descriptions with video content. GODIVA is capable of producing diverse and realistic video outputs from natural language descriptions, demonstrating the potential of transformers in bridging the gap between textual and visual modalities.

Building on this foundation, CogVideo (Hong et al., 2022) enhances the text-to-video synthesis process through large-scale pretraining. By leveraging vast amounts of data during pretraining, CogVideo improves the model's ability to generate coherent videos from a wide range of text inputs, making it more robust and versatile in handling different types of textual descriptions.

NÜWA (Wu et al., 2022) introduces a visual synthesis pretraining approach to further enrich the model's understanding of visual information before fine-tuning on text-to-video tasks. This pretraining phase allows the model to develop a more nuanced understanding of visual data, resulting in more accurate and realistic video generation based on textual input. NUWA-Infinity (Liang et al., 2022) extends the capabilities of its predecessor by enabling the generation of videos of arbitrary length. By stacking multiple autoregressive models, NUWA-Infinity overcomes the limitations of fixed-length video generation, providing greater flexibility and adaptability.

Phenaki (Villegas et al., 2022) takes a different approach by introducing variable-length video generation, allowing the model to produce videos that match the length and detail specified by the input text. This flexibility makes Phenaki particularly suitable for generating videos that need to accommodate varying levels of detail and duration. ART-V (Weng et al., 2024) introduces a masked diffusion model (MDM) to reduce drifting by determining which parts of the frame should rely on reference images versus network predictions. Additionally, anchored conditioning maintains consistency across long videos, allowing ART-V to produce detailed and aesthetically pleasing videos even with limited training resources. ViD-GPT (Gao et al., 2024) applies a GPT-style autoregressive framework within video diffusion models to generate high-fidelity video frames one at a time, conditioned on previously generated frames. By leveraging a latent space diffusion process and sophisticated temporal conditioning, ViD-GPT effectively reduces temporal inconsistencies and visual drift, resulting in videos that are both visually appealing and temporally coherent. Loong (Wang et al., 2024e) showcases capabilities in generating lengthy videos using a short-to-long video training strategy. PAV (Xie et al., 2024a) propose a progressive diffusion approach, improving both speed and fidelity in text-conditioned video generation. ARLON (Li et al., 2024f) integrates autoregressive models with diffusion transformers to provide long-range temporal guidance. LWM (Liu et al., 2024c) introduces an architecture capable of handling very long video sequences by leveraging Blockwise RingAttention. iVideoGPT (Wu et al.,

2024c) incorporates interactive mechanisms in video generation, making it more adaptable and scalable. Pandora (Xiang et al., 2024) uses a hybrid autoregressive-diffusion model to allow real time text controlled video generation.

**Visual Conditional Video Generation.**  Visual conditional video generation involves generating future video frames based on an initial sequence of frames or images. This task requires models to accurately capture and predict the temporal dynamics of video sequences. It relates to unconditional video generation in that the methods developed for unconditional video generation can also be applied to video prediction, providing a foundation for predicting future frames based on learned temporal patterns. The Convolutional LSTM Network (Shi et al., 2015) is one of the foundational models in this area, combining CNNs with LSTM units. This combination allows the model to effectively capture both spatial and temporal dependencies, making it particularly effective for tasks like precipitation nowcasting, where predicting the movement and evolution of visual patterns is crucial. Building upon this, PredRNN (Wang et al., 2017) proposes to use a spatiotemporal memory, and E3D-LSTM (Wang et al., 2019) integrates 3D convolutions into LSTM.

SV2P (Babaeizadeh et al., 2018) introduces stochastic elements into the video prediction process, allowing the model to account for multiple possible futures rather than predicting a single deterministic outcome. By learning a distribution over potential future frames, this model improves the realism and diversity of the generated videos, making it more adaptable to real-world scenarios where uncertainty and variability are inherent. PVV (Walker et al., 2021) and HARP (Seo et al., 2022) take advantage of latent space prediction to improve the accuracy and efficiency of video prediction. VQ-VAE compresses video data into a discrete latent space, where autoregressive models can more easily predict future frames. HARP builds on this approach by combining latent space prediction with a high-fidelity image generator, resulting in video sequences that are not only accurate but also visually detailed and temporally coherent. MaskViT enhances video prediction by using masked visual pretraining, where portions of video frames are masked during training, forcing the model to learn to predict the missing information. This approach improves the model's ability to handle occlusions and other challenging scenarios in video prediction tasks.

LVM (Bai et al., 2024a) introduces an approach that enhances the scalability of large vision models, making them more adaptable for multimodal tasks like image-to-video generation. ST-LLM (Liu et al., 2025) demonstrates that large language models are highly effective temporal learners, significantly advancing video generation tasks that rely on visual cues. Furthermore, Pyramid Flow (Jin et al., 2024a) introduces an efficient approach to flow matching, improving the computational efficiency of video generation models without compromising the temporal or spatial consistency of the generated videos. Building on the power of masked auto-regression, MarDini (Liu et al., 2024e) combines Masked AR with diffusion for scalable video generation.

**Multimodal Conditional Video Generation.**  Multimodal conditional video generation combines visual and textual inputs to create video sequences. MAGE (Hu et al., 2022) presents a key example of this approach, allowing users to control the motion and content of the generated video through text descriptions. This model provides precise control over the generated video, making it particularly useful for applications where detailed and specific video content is required, such as animation or visual storytelling.

While most video generation models focus on either unconditonal or conditional generation, there are models that aim to generalize across different types of video generation tasks, allowing a single model to generate videos from a wide range of inputs without requiring task-specific training. VideoPoet (Kondratyuk et al., 2024) is a significant step in this direction, leveraging a large language model (LLM) for zero-shot video generation. VideoPoet can generate videos from diverse inputs, including text, images, and existing videos, demonstrating the potential of LLMs in creating versatile and adaptable video generation models. This approach shows promise for developing truly universal models that can

handle the complex and varied demands of real-world video generation applications. The scalability of multimodal video generation has also been improved by sequential modeling techniques.

Autoregressive models have significantly advanced video generation, offering new capabilities in both unconditional and conditional video tasks. These approaches continue to evolve, improving scalability, flexibility, and generalization, bringing us closer to universal models capable of addressing the diverse demands of real-world video generation applications.

### 3.2.3 Embodied AI

While the previous subsections have focused on the generation of video sequences as an end goal, the emerging field of embodied AI introduces an important application for video generation. In this context, video generation is not merely about creating visually appealing or realistic sequences; instead, it serves as a critical component in training and augmenting intelligent agents that interact with and navigate their environments. By providing synthetic yet realistic visual data, video generation techniques can significantly enhance the learning process for embodied AI systems, enabling them to better understand and respond to dynamic, real-world scenarios. The following part explores how video generation is being leveraged to empower embodied AI, bridging the gap between visual synthesis and intelligent interaction.

Learning general world models in visual domains remains a significant challenge for policy learning in embodied AI. One approach is to learn action-conditioned video prediction models (Oh et al., 2015; Kaiser et al., 2020), which predict future frames based on current observations and actions. Advanced model-based reinforcement learning (RL) algorithms (Hafner et al., 2020; 2021; 2023; Schrittwieser et al., 2020; Hansen et al., 2022) enhance efficiency and accuracy by leveraging latent imagination. For example, IRIS (Micheli et al., 2023) uses a discrete autoencoder and an autoregressive Transformer to model dynamics as a sequence learning problem, improving sample efficiency with minimal tuning, particularly in the Atari 100k benchmark(Kaiser et al., 2020). However, these methods often complicate the process by tightly coupling model learning with policy learning.

To address this issue, recent efforts have focused on building world models that accumulate generalizable knowledge beyond specific tasks. Leveraging scalable architectures such as Transformers (Micheli et al., 2023) and pre-training on large-scale datasets (Wu et al., 2023a; Mendonca et al., 2023) have shown promise. GAIA-1 (Hu et al., 2023) integrates an autoregressive world model with a video diffusion decoder to predict future driving scenarios. By discretizing video frames into tokens, GAIA-1 trains a self-supervised autoregressive Transformer to predict the next frame at the representation level. These representations are subsequently decoded via a video diffusion process, yielding realistic frames with fine-grained details. iVideoGPT (Wu et al., 2024d) adopts a generic autoregressive Transformer framework to enhance the flexibility of scalable world models. Genie (Bruce et al., 2024) pre-trains world models by learning latent actions from videos without ground-truth actions, enabling action-controllable virtual worlds. GR-1 (Wu et al., 2024b) and GR-2 (Cheang et al., 2024) introduce GPT-style models for multi-task, language-conditioned visual robot manipulation. GR-1 is pretrained on video datasets and fine-tuned on robot data for end-to-end prediction of actions and future keyframes. GR-2, pretrained on 38 million video clips, generalizes across diverse robotic tasks and environments. Their architectures incorporate large-scale video generative pre-training into robotic manipulation tasks, thereby enhancing few-shot learning and generalization capabilities.

### 3.3 3D Generation

With increasing demand for realistic and controllable 3D content across various fields—from gaming and film to medical imaging and autonomous driving—researchers have been striving to advance 3D generation methods. Recent approaches leverage autoregressive models to improve control and detail in generated 3D structures. This section provides a comprehensive overview of key 3D generation advancements, focusing on the areas of motion, point clouds, scenes, and medical imaging. By exploring

these advancements, we observe the transformative potential of autoregressive methods in achieving more accurate and versatile 3D models.

### 3.3.1 Motion Generation

T2M-GPT (Zhang et al., 2023a) leverages pretrained text embeddings to generate detailed human motion sequences from textual descriptions. HiT-DVAE focuses (Bie et al., 2022) on generating 3D human motion sequences, utilizing a dynamical variational autoencoder (DVAE) to encode temporal dependencies in pose sequences into time-dependent latent variables, which are decoded across multiple scales to produce realistic motion. HuMoR (Rempe et al., 2021) models motion as a series of hierarchical latent variables, ensuring that both global movement patterns and finer details are accurately captured to produce realistic 3D motion sequences. AMD (Han et al., 2024) proposes a novel autoregressive model that iteratively generates complex 3D human motions from long text descriptions, leveraging the previous time-step text and motion sequences for improved temporal coherence and diversity.

### 3.3.2 Point Cloud Generation

Recent advances in autoregressive models have greatly enhanced 3D generation tasks. Canonical-VAE (Cheng et al., 2022) propose a transformer-based model for point cloud generation, decomposing point clouds into semantically aligned shape compositions, improving both reconstruction and unconditional generation. Octree Transformer (Ibing et al., 2023) introduces an octree-based method with adaptive compression, linearizing complex 3D data for efficient autoregressive generation. (Qian et al., 2024) present the Improved Auto-regressive Model (ImAM) for 3D shape generation, leveraging discrete representation learning for efficiency and supporting conditional generation. Additionally, Argus3D (Qian et al., 2024) scales these autoregressive models with 3.6 billion parameters, achieving impressive results on large-scale datasets.

### 3.3.3 Scene Generation

Inspired by the success of multi-scale alignment of visual and language tokens, more tasks are adopting similar approaches.SceneScript (Avetisyan et al., 2024) proposes a novel autoregressive image generation approach by using structured language commands to represent scenes. This method provides compact and interpretable scene representations, excelling in architectural layout estimation and 3D object detection, while being easily extendable to new tasks.

### 3.3.4 3D Medical Generation

3D images are challenging to acquire due to cost, quality, and accessibility constraints. Aiming to produce high-resolution 3D volumetric imaging data with the correct anatomical morphology, SynthAnatomy (Tudosiu et al., 2022) and BrainSynth (Tudosiu et al., 2024) scale and optimize VQ-VAE and Transformer models for high-resolution volumetric data. Similarly, ConGe (Zhou & Khalvati, 2024) and 3D-VQGAN (Zhou et al., 2023b) introduce the class-conditional generation framework for synthesizing 3D brain tumor MRI ROIs. Another way to generate 3D representations is image synthesis based on 2D images. To achieve this goal, Unalign (Corona-Figueroa et al., 2023) reformulate it as a voxel-to-voxel prediction problem and achieve it by conditioning an unconstrained transformer on 2D input views. Specifically, they model such mapping with a conditional likelihood-based generative model, allowing sampled 3D data to sit at arbitrary positions/rotations relative to the 2D data. Moreover, AutoSeq (Wang et al., 2024c) introduces an autoregressive pre-training framework to represent 3D medical images, which treats them as interconnected visual tokens, which performs well on several downstream tasks in public datasets, demonstrating the strong ability of autoregressive training.

In summary, autoregressive modeling techniques have significantly advanced the generation of complex and diverse 3D structures. By harnessing temporal dependencies, shape compositions, and conditional

representations, these methods enhance fidelity and offer refined control over generated outputs. The diverse applications across motion, point clouds, scene creation, and medical imaging underscore the versatility of autoregressive 3D generation. Future research is poised to further refine these models, pushing the boundaries of realism, scalability, and usability across various domains.

## 3.4 Multimodal Understanding and Generation

Previous sections discussed visual generation, which transforms language or visual input into visual output. In this section, we shift our focus to multimodal-to-multimodal tasks. This section is divided into two parts: the framework for multimodal understanding and the framework for unifying multimodal understanding and generation.

### 3.4.1 The Framework for Multimodal Understanding

The evolution of sophisticated models in multimodal understanding has significantly advanced the integration of visual and textual data. A key approach in this domain is the discrete image token masked image modeling (MIM) method, which has gained traction as a powerful pretraining strategy. This method is instrumental in learning robust visual representations by predicting missing parts of images, thus allowing models to gain a deeper understanding of visual content in context.

One pivotal model built on this groundwork is the BEiT (Bao et al., 2021). BEiT leverages MIM by treating images analogously to how BERT treats text, masking parts of the image and predicting the discrete tokens for these masked sections. This enables the model to learn high-level visual features efficiently, akin to how BERT captures linguistic nuances.

The lineage of BEiT has seen multiple significant enhancements. BEiT-v2 (Peng et al., 2022) introduced the concept of visual quantization knowledge distillation (VQKD), which integrates semantic information into the image tokenization process. By leveraging pretrained models like CLIP, BEiT-v2 incorporates semantic knowledge into the token representations, thereby improving the model's ability to capture both visual and semantic features more effectively.

VL-BEiT (Bao et al., 2022) represents a further advancement in the series, focusing on generative vision-language pretraining. It employs a unified transformer-based architecture to seamlessly process and integrate both image and text data. By using a generative approach for pretraining, VL-BEiT is capable of learning joint visual and textual representations, enhancing its ability to perform tasks such as image captioning and visual question answering with high accuracy and coherence. BEiT-v3 (Wang et al., 2022b) continued to refine these ideas, integrating more sophisticated multimodal techniques and achieving state-of-the-art results in a variety of vision-language benchmarks.

Additionally, Flamingo (Alayrac et al., 2022) demonstrated that in-context learning can be effectively scaled to large-scale vision-language models. Specifically, they enhanced downstream tasks like image captioning by requiring only a few examples for significant improvements.

As the field progressed, new frameworks like LLaVA (Liu et al., 2024d) emerged, representing the latest wave in multimodal understanding. LLaVA is built upon the extensive capabilities of large language models, integrating them with powerful visual representation modules, thereby setting new paradigms in the seamless fusion of language and vision tasks.

Several works like the AIM series (El-Nouby et al., 2024; Fini et al., 2024) have also explored a compelling framework for learning visual representations through generative pre-training. In AIM (El-Nouby et al., 2024), images are decomposed into non-overlapping patches, and a causal Vision Transformer (ViT) (Dosovitskiy, 2020) is trained to autoregressively predict the next patch's representation. These predicted representations are then mapped to pixel space via a heavyweight MLP for pixel-level reconstruction. This approach enables AIM to learn robust visual representations using only a generative objective. AIMV2 Fini et al. (2024) further enhances this by aligning visual representations with

textual features, expanding its applicability to multimodal understanding tasks. Both models rely solely on generative pretraining and exhibit strong scaling properties, mirroring trends observed in LLMs Achiam et al. (2023); Dubey et al. (2024), and demonstrating that large autoregressive models can serve as a versatile paradigm for unified representation learning across modalities.

These advancements mark significant strides in bridging visual and textual data, facilitating richer, more context-aware AI systems that can process and interpret the world in a more human-like manner.

### 3.4.2 The Framework for Unifying Multimodal Understanding and Generation

The unified framework extends traditional multimodal understanding (Chen et al., 2022; Li et al., 2023a; Zhang et al., 2023b; Shi et al., 2023; Xie et al., 2024b; Wan et al., 2024) by enabling both visual and textual output generation. Unlike earlier models that focused solely on interpretation, this framework leverages the integration of Large Language Models (LLMs) to enhance multimodal content generation. The overview of auto-regressive multimodal LLM's application is shown in Figure 10.

From pre-LLM era's separate processing of modalities, the evolution to LLMs has allowed for a seamless convergence of language and visual data. This progress supports more coherent and contextually rich outputs, using native multimodal architectures to handle integrated inputs and outputs efficiently. As a result, the Unified Framework facilitates applications that require simultaneous comprehension and generation across modalities, such as interactive storytelling and dynamic content creation.

**Autoregressive Vision-Language Fusion Methods.** Before the advent of LLMs, several autoregressive models focused on integrating vision and language. Notable among these is OFA (Wang et al., 2022a), which unified tasks across modalities using a shared transformer backbone, demonstrating versatility in handling tasks such as image captioning and visual question answering. Similarly, CogView (Ding et al., 2021) and M6 (Lin et al., 2021), extended the capabilities to Chinese text inputs, offering robust image generation from textual descriptions. ERNIE-ViLG (Zhang et al., 2021) further advanced this by employing a multi-modal pretraining strategy that enhanced the understanding of image-text pairs. Unified-IO (Lu et al., 2022) explored input-output transformations across modalities, providing a unified framework for diverse tasks.

**Integration with LLMs and Diffusion Models.** With the rise of LLMs, autoregressive models have increasingly interfaced with diffusion models to enhance multi-modal capabilities. NEXT-GPT (Wu et al., 2023c) exemplifies this trend by combining the autoregressive approach with diffusion processes to improve image synthesis from textual inputs. The SEED (Ge et al., 2024) and EMU-series (Sun et al., 2024b; 2023; Wang et al., 2024d) further explore this synergy, utilizing diffusion models to refine and stabilize the output quality of generated visual content. LaViT (Jin et al.) employs a token merging strategy, which reduces the number of visual tokens by merging tokens that are spatially or semantically similar, thereby lowering computational costs while maintaining high-quality visual generation. Video-LaViT (Jin et al., 2024c), an extension of LaViT, applies a keyframe-based token reduction technique to video data, selecting keyframes that capture the most critical information and sparsely sampling intermediate frames, significantly reducing token count while preserving temporal coherence in video understanding and generation. Additionally, models like X-ViLA (Ye et al., 2024) (Cross-modality Vision, Language, Audio) push the boundaries of multi-modal learning by serving as a foundation model for cross-modality understanding, reasoning, and generation across video, image, language, and audio domains. X-ViLA demonstrates the potential of unified multi-modal architectures to handle diverse tasks across multiple data types, offering a holistic approach to multi-modal content generation. This integration has proven beneficial in addressing the limitations of purely autoregressive methods, particularly in generating high-fidelity and diverse images.

**Autoregressive Models for Seamless Multi-Modal Integration.** More recent developments have seen the emergence of native multi-modal autoregressive models designed from the ground up to

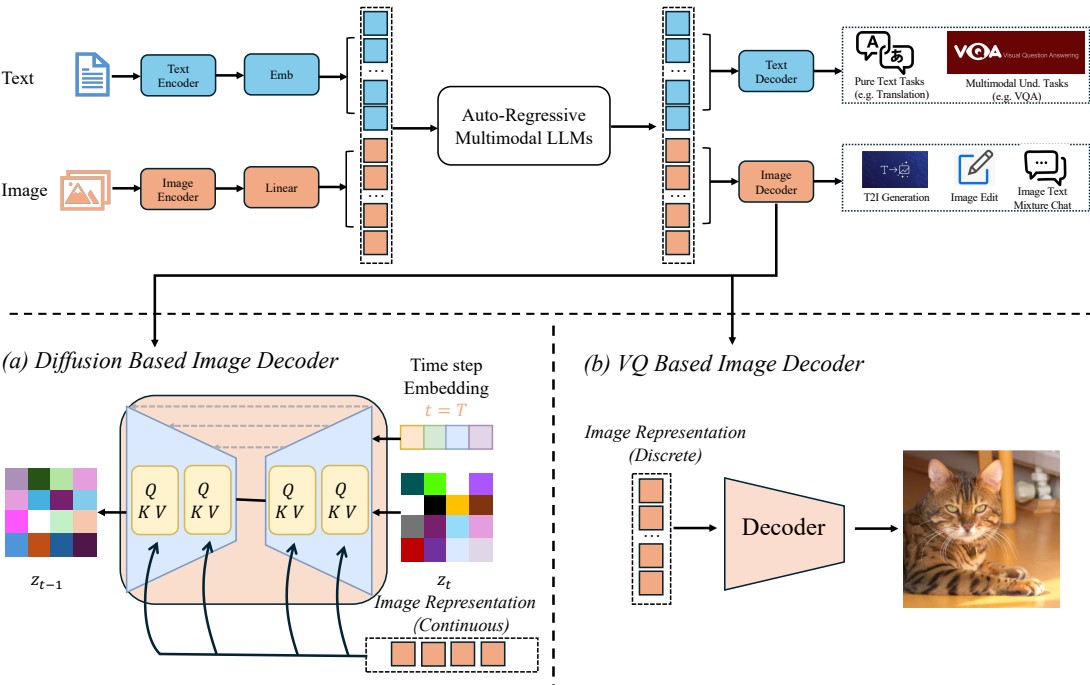

Figure 10: **The overview of auto-regressive multi-modal LLMs' application.** The diagram highlights a high-level architecture combining both text and image modalities within auto-regressive LLMs. The image and text encoders process input data, which is subsequently integrated by the multimodal model to generate text and image outputs. Two image decoding strategies are shown: (a) Diffusion-based image decoder, where continuous image representations are progressively refined using a transformer-based diffusion process, and (b) VQ-based image decoder, where discrete image tokens are transformed into visual outputs. The framework is capable of handling a range of tasks, including pure text-based tasks (e.g., translation), vision-language tasks (e.g., VQA), text-to-image generation, image editing, and multi-modal conversational applications.

handle multiple data types. Models such as Chameleon (Team, 2024) and Transfusion (Zhou et al., 2024) exemplify this new class of architectures, providing smooth integration of visual and textual data. These models are built to natively support multi-modal tasks, offering enhanced flexibility and scalability. Chameleon, for instance, is designed to handle multi-modal inputs with minimal architectural adjustments, making it a versatile model for tasks ranging from image captioning to multi-modal reasoning.

Aghajanyan et al. (2023) is an important precursor to these developments, exploring how scaling laws apply to generative models that handle multiple modalities simultaneously. This work provides critical insights into how model size, data quantity, and computation scale influence performance in mixed-modal settings, guiding the design of larger, more capable multi-modal models like Chameleon and Transfusion. Another notable model is RA-CM3 (Yasunaga et al., 2022), which incorporates retrieval-augmented techniques to enhance multi-modal language modeling. By integrating a retrieval mechanism, RA-CM3 improves the model's ability to access relevant external knowledge, thereby boosting performance on tasks requiring a deeper understanding of both visual and textual inputs. This retrieval-augmented approach demonstrates how external knowledge sources can be leveraged to enhance the capabilities of multi-modal models, particularly in tasks that require complex reasoning or contextual understanding. SHOW-o (Xie et al., 2024b), for instance, introduces an innovative modeling scheme that aligns image and text representations, facilitating tasks such as image generation

and visual storytelling, with the use of MAGVIT-v2 (Yu et al., 2023b) tokenizer. VILA-U (Wu et al., 2024e) injects high-level visual information into RQ-VAE tokenizer (Lee et al., 2022a) from a pretrained CLIP (Radford et al., 2021) to unify understanding and generation. Janus (Wu et al., 2024a) explicitly decouples the understanding and generation vision encoder, which avoids the conflict between vision understanding and generation.

To further enhance the efficacy of visual in-context learning, CoTVL (Ge et al., 2023) successfully applies visual chain-of-thought prompt tuning for vision-language modeling, and performs better in tasks that require more reasoning abilities. These models highlight the potential for autoregressive frameworks to serve as foundational architectures in multi-modal applications, promoting a more holistic approach to understanding and generating content across modalities.

The previously mentioned multimodal tokenization involves encoding images into tokens, allowing Large Language Models (LLMs) to process both visual and language signals in a unified space. Vision-to-Language (V2T) Tokenizer (Zhu et al., 2024a) maps image patches to discrete tokens that correspond to LLM vocabularies, enabling tasks like inpainting and deblurring. Similarly, the Multimodal Cross-Quantization VAE (MXQ-VAE) (Lee et al., 2022b) encodes both image and text inputs as tokens, generating coherent multimodal outputs. These approaches improve image-text generation, though they typically predict pixels in a fixed order without considering random generation strategies.

In summary, the evolution of visual autoregressive models in the realm of multi-modality underscores their growing importance and versatility. From early integrations to sophisticated native architectures, these models continue to push the boundaries of what's possible in generating and understanding multi-modal content. Future research directions may focus on improving model efficiency, scalability, and the ability to handle an even broader array of modalities.

## 4 Evaluation Metrics

This section presents commonly used quantitative and qualitative metrics to evaluate the visual autoregressive models, summarized in Table 2.

### 4.1 Evaluation of Visual Tokenizer Reconstruction

The Visual Tokenizer serves a crucial role in compressing visual content into discrete token sequences and accurately reconstructing it. Evaluation metrics for these tokenizers primarily focus on reconstruction fidelity, with widely adopted metrics including PSNR, SSIM (Wang et al., 2004), LPIPS (Zhang et al., 2018), and rFID (Heusel et al., 2017). PSNR (Peak Signal-to-Noise Ratio) and SSIM (Structural Similarity Index Measure) are pixel-level metrics that quantify the degree of pixel-wise alignment between the reconstructed image and its reference. While PSNR captures the average signal-to-noise ratio across image pixels, SSIM accounts for perceptual differences in structure, luminance, and contrast. LPIPS, in contrast, measures perceptual similarity by evaluating features from a pretrained VGG network (Simonyan & Zisserman, 2014). rFID (Reconstruction Fréchet Inception Distance) extends the traditional FID to the reconstruction setting, measuring the distributional discrepancy between original and reconstructed image sets in feature space.

### 4.2 Evaluation of Visual Autoregressive Generation

Beyond the evaluation of tokenizers, comprehensive assessments of visual autoregressive generation models are essential to measure their performance across multiple dimensions. Key metrics include the following five aspects:

1. **Visual Quality** assesses the realism of generated content relative to real data. One fundamental metric is Negative Log-Likelihood (NLL), which directly quantifies the likelihood of generated data under the model's learned distribution. Additional metrics, such as Inception Score (IS) (Salimans

Table 2: Evaluation metrics used to assess Visual autoregressive models performance. FR: Full-Reference, NR: No-Reference. ↑ indicates that the higher the metric the better the model performance and vice versa.

| Metric | Reference Dependency | Source |
|---|---|---|
| **Reconstruction Fidelity** | | |
| Peak Signal-to-Noise Ratio (PSNR) ↑ | FR | - |
| Structural Similarity Index Measure (SSIM) ↑ | FR | Wang et al. (2004) |
| Learned Perceptual Image Patch Similarity (LPIPS) ↓ | FR | Zhang et al. (2018) |
| Reconstruction Fréchet Inception Distance (rFID) ↓ | FR | Heusel et al. (2017) |
| **Visual Quality** | | |
| Negative Log-Likelihood (NLL) ↓ | NR | Fisher (1922) |
| Inception Score (IS) ↑ | NR | Salimans et al. (2016) |
| Fréchet Inception Distance (FID) ↓ | NR | Heusel et al. (2017) |
| Kernel Inception Distance (KID) ↓ | NR | Bińkowski et al. (2018) |
| Fréchet Video Distance (FVD) ↓ | NR | Unterthiner et al. (2018) |
| Aesthetic Score ↑ | NR | Schuhmann (2022) |
| **Diversity** | | |
| Precision and Recall ↑ | NR | Kynkäänniemi et al. (2019) |
| MODE Score ↑ | NR | Che et al. (2022) |
| **Semantic Consistency** | | |
| CLIP Similarity (CLIPSIM) ↑ | NR | Radford et al. (2021) |
| R-precision ↑ | NR | Craswell (2009) |
| **Temporal Coherence** | | |
| Warping Errors ↑ | NR | Lai et al. (2018) |
| CLIP Similarity Among Frames (CLIPSIM-Temp) ↑ | NR | Radford et al. (2021) |
| **Human-Centered Assessment** | | |
| Human Preference Score (HPS) ↑ | NR | Wu et al. (2023d) |
| Quality ELO Score ↑ | NR | Elo (1978) |
| Human Study Metrics (Ratings, Preference, etc.) | NR | - |

et al., 2016), Fréchet Inception Distance (FID) (Heusel et al., 2017), and Kernel Inception Distance (KID) (Bińkowski et al., 2018), measure the distributional closeness between generated samples and real data. which evaluates different aspects of the generated content's realism and closeness to real data distributions. Aesthetic Score (Schuhmann, 2022), a CLIP-based method, also contributes to this evaluation by measuring the aesthetic appeal of generated images through embeddings aligned with human aesthetic preferences. For video generation tasks, the Fréchet Video Distance (FVD) (Unterthiner et al., 2018) extends FID by taking temporal modeling into consideration, evaluating the distance between distributions of generated and real videos.

2. **Diversity** evaluates the variety and richness of generated outputs within the model's distribution. Precision and Recall (Kynkäänniemi et al., 2019) metrics here assess both fidelity and diversity: Precision measures the proportion of generated samples that lie within the real data distribution, while Recall evaluates the extent to which the real data distribution is represented within the generated outputs. While the MODE Score (Che et al., 2022) quantifies both the quality and diversity of the generated samples.

3. **Semantic Consistency** examines the alignment between generated content and given textual descriptions or other modalities of input. Metrics such as CLIP Score and R-precision (Craswell, 2009) are commonly used to assess this alignment. The CLIP Score quantifies the cosine similarity

| Type | Model | #Params | Resolution | FID ↓ | IS ↑ | Precision ↑ | Recall ↑ |
|---|---|---|---|---|---|---|---|
| **GAN** | BigGan (Brock, 2018) | 112M | 256 × 256 | 6.95 | 224.50 | 0.89 | 0.38 |
| | GigaGan (Kang et al., 2023) | 569M | 256 × 256 | 3.45 | 225.50 | 0.84 | 0.61 |
| | StyleGan-XL (Sauer et al., 2022) | 166M | 256 × 256 | 2.30 | 265.10 | 0.78 | 0.53 |
| **Diffusion** | ADM (Dhariwal & Nichol, 2021) | 554M | 256 × 256 | 10.94 | 101.00 | 0.69 | 0.63 |
| | CDM (Daras et al., 2024) | - | 256 × 256 | 4.88 | 158.70 | - | - |
| | LDM-4 (Rombach et al., 2022) | 400M | 256 × 256 | 3.60 | 247.70 | - | - |
| | DiT-XL/2-G (Peebles & Xie, 2023) | 675M | 256 × 256 | 2.27 | 278.20 | 0.83 | 0.57 |
| | VDM++ (Kingma & Gao, 2024) | 2B | 256 × 256 | 2.40 | 225.30 | - | - |
| | DiT-XL/2-G (Peebles & Xie, 2023) | 675M | 512 × 512 | 3.04 | 240.82 | 0.84 | 0.54 |
| **Mask** | MaskGIT (Chang et al., 2022) | 227M | 256 × 256 | 6.18 | 182.10 | 0.80 | 0.51 |
| | MaskGIT -re (Chang et al., 2022) | 227M | 256 × 256 | 4.02 | 355.60 | - | - |
| **AR** | VQGAN -re (Esser et al., 2021b) | 1.4B | 256 × 256 | 5.20 | 280.30 | - | - |
| | VQGAN (Esser et al., 2021b) | 227M | 256 × 256 | 18.65 | 80.40 | 0.78 | 0.26 |
| | VQGAN (Esser et al., 2021b) | 1.4B | 256 × 256 | 15.78 | 74.30 | - | - |
| | RQTran. -re (Lee et al., 2022a) | 1.7B | 256 × 256 | 3.80 | 323.70 | - | - |
| | RQTran. (Lee et al., 2022a) | 1.7B | 256 × 256 | 7.55 | 134.00 | - | - |
| | ViT-VQGAN -re (Yu et al., 2021a) | 1.7B | 256 × 256 | 3.04 | 227.40 | - | - |
| | ViT-VQGAN (Yu et al., 2021a) | 3.8B | 256 × 256 | 4.17 | 175.10 | - | - |
| | LlamaGen-B (Sun et al., 2024a) | 111M | 256 × 256 | 5.46 | 193.61 | 0.83 | 0.45 |
| | LlamaGen-L (Sun et al., 2024a) | 343M | 256 × 256 | 3.07 | 256.06 | 0.83 | 0.52 |
| | LlamaGen-XL (Sun et al., 2024a) | 775M | 256 × 256 | 2.62 | 244.08 | 0.80 | 0.57 |
| | LlamaGen-XXL (Sun et al., 2024a) | 1.4B | 256 × 256 | 2.34 | 253.90 | 0.80 | 0.59 |
| | LlamaGen-3B (Sun et al., 2024a) | 3B | 256 × 256 | 2.18 | 263.33 | 0.81 | 0.58 |
| | Open-MAGVIT2-B (Luo et al., 2024) | 343M | 256 × 256 | 3.08 | 258.26 | 0.83 | 0.52 |
| | Open-MAGVIT2-L (Luo et al., 2024) | 804M | 256 × 256 | 2.51 | 271.70 | 0.84 | 0.54 |
| | Open-MAGVIT2-XL (Luo et al., 2024) | 1.5B | 256 × 256 | 2.33 | 271.77 | 0.84 | 0.54 |
| | VAR-d16 (Tian et al., 2024) | 310M | 256 × 256 | 3.30 | 274.40 | 0.84 | 0.51 |
| | VAR-d20 (Tian et al., 2024) | 600M | 256 × 256 | 2.57 | 302.60 | 0.83 | 0.56 |
| | VAR-d24 (Tian et al., 2024) | 1.0B | 256 × 256 | 2.09 | 312.90 | 0.82 | 0.59 |
| | VAR-d30 (Tian et al., 2024) | 2.0B | 256 × 256 | 1.92 | 323.10 | 0.82 | 0.59 |
| | SPAE (Yu et al., 2024a) | - | 128 × 128 | 4.41 | 133.03 | - | - |
| | SPAE (Yu et al., 2024a) | - | 256 × 256 | 3.60 | 168.50 | - | - |
| | MAR-B (Li et al., 2024c) | 208M | 256 × 256 | 3.48 | 192.40 | 0.78 | 0.58 |
| | MAR-L (Li et al., 2024c) | 479M | 256 × 256 | 2.60 | 221.40 | 0.79 | 0.60 |
| | MAR-H (Li et al., 2024c) | 943M | 256 × 256 | 2.35 | 227.80 | 0.79 | 0.62 |
| | MAR-H w/ CFG (Li et al., 2024c) | 943M | 256 × 256 | 1.55 | 303.70 | 0.81 | 0.62 |
| | DART w/ CFG (Zhao et al., 2024) | 812M | 256 × 256 | 3.98 | - | - | - |

Table 3: Comparison of Model Parameters, Resolution, FID, IS, Precision, and Recall across various Types of Generative Models on ImageNet dataset (Deng et al., 2009). -re is the generative model with rejection sampling. w/ CFG is the model with classifier-free diffusion guidance (Ho & Salimans, 2022).

between visual outputs and textual input using a pretrained CLIP (Radford et al., 2021) model, while R-precision measures the ranking of relevant generated samples with respect to their alignment with the intended context.

4. **Temporal Coherence** focuses on the temporal consistency across generated frames. Warping Errors (Lai et al., 2018), based on optical flow (Teed & Deng, 2020), measure the consistency of motion and object placement over time. The additional commonly applied metric is CLIPSIM-Temp, which calculates the CLIP similarity between successive frame embeddings.

5. **Human-Centered Assessment** is crucial for evaluating the subjective quality of generated visual content, while quantitative metrics provide essential objective assessments. The Human Preference Score (HPS) (Wu et al., 2023d) leverages a classifier trained on extensive human preference data, estimating how closely generated outputs align with human aesthetic judgments. Recently, the Quality ELO Score (Elo, 1978) has gained prominence: in this method, generative models compete on an online platform (Analysis, 2023) where users express preferences between paired model outputs. These preferences are then used to compute ELO scores, effectively capturing human rankings of models.

| Type | Model | #Para. | MJHQ-30K FID ↓ | MJHQ-30K CLIP-Score ↑ | MS-COCO FID-30K ↓ |
|---|---|---|---|---|---|
| **Diffusion** | LDM (Rombach et al., 2022) | 1.4B | 12.64 | - | 12.64 |
| | DALL-E-2 (Ramesh et al., 2022) | 6.5B | 10.39 | - | 10.39 |
| | Imagen (Saharia et al., 2022) | 3B | 7.27 | - | 7.27 |
| | SD2.1 (Rombach et al., 2022) | 860M | 26.96 | 25.90 | - |
| | SD3-Medium (Esser et al., 2024) | 2B | 11.92 | 28.83 | - |
| | SDXL (Esser et al., 2024) | 2.6B | 8.76 | 28.60 | - |
| | PixArt-Σ (Chen et al., 2024) | 630M | 6.34 | 27.62 | - |
| | PixArt-α (Chen et al., 2023) | 630M | 6.14 | 27.55 | - |
| | Playground v2.5 (Li et al., 2024a) | 2B | 6.84 | 29.39 | - |
| | RAPHAEL (Xue et al., 2024) | 3B | - | - | 6.61 |
| **AR (Gen)** | DALLE (Ramesh et al., 2021) | 12B | - | - | 27.50 |
| | Fluid (Fan et al., 2024) | 10.5B | 6.16 | - | 6.16 |
| | Fluid (Fan et al., 2024) | 3.1B | 6.41 | - | 6.41 |
| | Fluid (Fan et al., 2024) | 1.1B | 6.59 | - | 6.59 |
| | Fluid (Fan et al., 2024) | 665M | 6.84 | - | 6.84 |
| | Fluid (Fan et al., 2024) | 369M | 7.23 | - | 7.23 |
| | Muse (Chang et al., 2023) | 3B | 7.88 | - | 7.88 |
| | Parti (Yu et al., 2022) | 20B | 7.23 | - | 7.23 |
| | Emu3-Gen (Wang et al., 2024d) | 8B | 25.59 | - | - |
| | LlamaGen (Sun et al., 2024a) | 775M | 25.59 | 23.03 | - |
| | HART (Tang et al., 2024) | 732M | 5.22 | 29.01 | - |
| | DART (Zhao et al., 2024) | 812M | - | - | 11.12 |
| **AR (Unified)** | Seed-X (Ge et al., 2024) | 17B | 10.82 | - | 14.99 |
| | LVM (Bai et al., 2024b) | 7B | 17.77 | - | 12.68 |
| | Chameleon (Team, 2024) | 34B | - | - | 10.82 |
| | Show-o (Xie et al., 2024b) | 1.3B | 14.99 | 27.02 | - |
| | VILA-U (Wu et al., 2024e) | 7B | 12.81 | - | - |
| | Janus (Wu et al., 2024a) | 1.3B | 10.10 | - | 8.53 |
| | Transfusion (Zhou et al., 2024) | 7.3B | 6.61 | - | 6.61 |

Table 4: Comparison of Model Parameters, FID, CLIP-Score across various Types of Generative Models on MJHQ-30K (Li et al., 2024a) and MS-COCO (Lin et al., 2014) datasets.

| Type | Model | #Para. | GenEval ↑ Single Obj. | Two Obj. | Counting | Color | Position | Color Attri. | Overall |
|---|---|---|---|---|---|---|---|---|---|
| **Diffusion** | LDM (Rombach et al., 2022) | 1.4B | 0.92 | 0.29 | 0.23 | 0.70 | 0.02 | 0.05 | 0.37 |
| | DALL-E-2 (Ramesh et al., 2022) | 6.5B | 0.94 | 0.66 | 0.49 | 0.77 | 0.10 | 0.19 | 0.52 |
| | DALL-E-3 (Betker et al., 2023) | - | 0.96 | 0.87 | 0.47 | 0.83 | 0.43 | 0.45 | 0.67 |
| | SD1.5 (Rombach et al., 2022) | 860M | 0.97 | 0.38 | 0.35 | 0.76 | 0.04 | 0.06 | 0.43 |
| | SD2.1 (Rombach et al., 2022) | 860M | 0.98 | 0.51 | 0.44 | 0.85 | 0.07 | 0.17 | 0.55 |
| | SD3-Large (Esser et al., 2024) | 8B | 0.98 | 0.84 | 0.66 | 0.74 | 0.40 | 0.43 | 0.68 |
| | SDXL (Esser et al., 2024) | 2.6B | 0.98 | 0.74 | 0.39 | 0.85 | 0.15 | 0.23 | 0.58 |
| | PixArt-α (Chen et al., 2023) | 630M | 0.98 | 0.50 | 0.44 | 0.80 | 0.08 | 0.07 | 0.48 |
| **AR (Gen)** | Fluid (Fan et al., 2024) | 10.5B | 0.96 | 0.83 | 0.63 | 0.80 | 0.39 | 0.51 | 0.69 |
| | Fluid (Fan et al., 2024) | 3.1B | 0.83 | 0.83 | 0.60 | 0.82 | 0.41 | 0.53 | 0.70 |
| | Fluid (Fan et al., 2024) | 1.1B | 0.96 | 0.77 | 0.61 | 0.78 | 0.34 | 0.53 | 0.67 |
| | Fluid (Fan et al., 2024) | 665M | 0.96 | 0.73 | 0.51 | 0.77 | 0.42 | 0.51 | 0.65 |
| | Fluid (Fan et al., 2024) | 369M | 0.96 | 0.64 | 0.53 | 0.78 | 0.33 | 0.46 | 0.62 |
| | Emu3-Gen (Wang et al., 2024d) | 8B | 0.98 | 0.71 | 0.34 | 0.81 | 0.17 | 0.21 | 0.54 |
| | LlamaGen (Sun et al., 2024a) | 775M | 0.71 | 0.34 | 0.21 | 0.58 | 0.07 | 0.04 | 0.32 |
| **AR (Unified)** | Seed-X (Ge et al., 2024) | 17B | 0.97 | 0.58 | 0.26 | 0.80 | 0.19 | 0.14 | 0.49 |
| | LVM (Bai et al., 2024b) | 7B | 0.93 | 0.41 | 0.46 | 0.79 | 0.09 | 0.15 | 0.48 |
| | Chameleon (Team, 2024) | 34B | - | - | - | - | - | - | 0.39 |
| | Show-o (Xie et al., 2024b) | 1.3B | 0.95 | 0.52 | 0.49 | 0.82 | 0.11 | 0.28 | 0.53 |
| | Janus (Wu et al., 2024a) | 1.3B | 0.97 | 0.68 | 0.30 | 0.84 | 0.46 | 0.42 | 0.61 |
| | Transfusion (Zhou et al., 2024) | 7.3B | - | - | - | - | - | - | 0.63 |

Table 5: Comparison of various types of generative models on GenEval bench (Ghosh et al., 2024).

Additionally, there are various User Study Metrics, such as ratings, preferences, and other subjective measures collected from user studies. These metrics are often case-specific and provide valuable insights into evaluation.

### 4.3 Task-Specific Evaluation Metrics

In the previous sections, we categorized and introduced several common metrics based on various evaluation dimensions. In this subsection, we reorganize these metrics according to specific tasks. The aim is to provide task-specific classifications, allowing readers to quickly identify the most relevant metrics for each task. Specifically, metrics for visual autoregressive models are categorized as follows:

- **Reconstruction**
  - PSNR, SSIM (Wang et al., 2004), LPIPS (Zhang et al., 2018), rFID (Heusel et al., 2017) are pixel-level accuracy metrics for restoration tasks, which is commonly adopted for evaluating the performance of tokenizers in token-based autoregressive models. Additionally, codebook utilization is often employed to assess how effectively vectors in VQ codebooks are utilized.

- **Image Generation**
  - **General Metrics.** FID (Heusel et al., 2017), IS (Salimans et al., 2016), and KID (Bińkowski et al., 2018) are widely adopted for evaluating the quality of generated images. Precision and Recall (Kynkäänniemi et al., 2019) are typically used to assess the fidelity and diversity of generated content.
  - **Text-to-Image Generation.** CLIPSim (Radford et al., 2021) and R-Precision (Craswell, 2009) are key metrics for evaluating the relevance between text and images in text-to-image generation tasks.
  - **Inpainting** In addition to the general quality metrics, pixel-level metrics are employed to assess whether the inpainting results align with the ground truth.

- **Video Generation**
  - **General Metrics.** FVD (Unterthiner et al., 2018) is commonly used to evaluate the video quality. CLIPSIM-Temp (Radford et al., 2021) and Wrapping Errors (Lai et al., 2018) are used to assess the temporal coherence across video generation tasks.
  - **Text-to-Video Generation.** Similar to T2I Generation, T2V generation uses semantic consistency metrics like CLIPSIM to evaluate alignment with given text descriptions. Benchmarks such as VBench (Huang et al., 2024) and EvalCrafter (Liu et al., 2024h) are also widely used to evaluate the overall performance of T2V generation.

- **Task-Specific Metrics**
  - In addition, there are various task-specific metrics and benchmarks tailored to particular tasks. These are designed to assess specified attributes according to task requirements. For example, T2I-CompBench (Huang et al., 2023a) is used for evaluating compositional generation tasks, while CLIPSIM-Img is applied to example-based image editing tasks.

### 4.4 Comparison of Existing Models

Based on the collected results presented in Table 3, Table 4, and Table 5, we conduct a comprehensive analysis of different model types and their performance characteristics. First, comparing AR and diffusion models reveals interesting patterns across metrics. While both approaches demonstrate comparable performance in Precision and Recall metrics, as well as FID scores, AR models show a notable advantage in IS metrics. For instance, in Table 3, AR models consistently achieve higher IS scores, with many models exceeding 250, while diffusion models typically show lower IS values. Second, the comparison between specialized generation models (AR Gen) and unified models (AR Unified) reveals clear performance differences across both Table 4 and Table 5. In Table 4, specialized generation models achieve better FID scores compared to unified models. This pattern continues in Table 5, where specialized generation models like Fluid demonstrate superior GenEval scores (0.70) compared

to unified models. These consistent results across different metrics and datasets suggest that current approaches to unifying understanding and generation might actually limit generation performance. While the initial hypothesis suggested that unified understanding and generation would enhance generation quality, our analysis indicates otherwise. This performance gap might be attributed to the current limitations in vision tokenization methods, which have not yet achieved a seamless integration of understanding and generation capabilities. The distinct requirements for visual understanding versus generation tasks may create competing objectives that current unified architectures struggle to optimize simultaneously. This observation suggests that further research is needed to develop more effective approaches for unifying visual understanding and generation. The consistent performance gap indicates that fundamental advances in vision tokenization and architectural design may be required to realize the theoretical benefits of unified models.

## 5 Challenges and Future Work

### 5.1 Technical challenges

**How to design a powerful tokenizer?** This paper categorizes existing autoregressive models into three primary types: next-pixel prediction, next-token prediction and next-scale prediction. Among these, next-token prediction and next-scale prediction have gained prominence. Both approaches hinge critically on the availability of a powerful tokenizer that can effectively compress images or videos into discrete visual tokens. However, designing such a tokenizer is a non-trivial challenge. One of the most widely adopted methods is VQGAN (Esser et al., 2021b), which employs VQ techniques and adversarial training to develop a perceptually rich tokenizer. Vanilla VQGAN adopts a relatively large vector dimension(256), allowing each vector to represent rich semantic information. However, as the codebook size scales, this capacity becomes a bottleneck, significantly limiting the utilization rate of larger codebooks, and presenting challenges for applying VQGAN in large-scale autoregressive models. A rising trend in codebook design (Sun et al., 2024a) involves using smaller vector dimensions with larger codebook sizes to enhance lookup efficiency and codebook utilization. Some studies (Yu et al., 2023b; Luo et al., 2024) have even pushed the boundaries by reducing the vector dimension to zero, developing a binary lookup-free quantization(LFQ) with a codebook size of $2^{18}$. Additionally, various advanced training strategies (Razavi et al., 2019; Yu et al., 2021a; Zhu et al., 2024b) have been explored to further improve VQGAN's codebook utilization. Another promising avenue is leveraging hierarchical multi-scale properties (Lee et al., 2022a; Tian et al., 2024) to improve the compression of visual data. In summary, designing a powerful tokenizer for open-domain visual data is crucial for advancing autoregressive visual generation.

**Discrete or continuous?** Autoregressive models have traditionally been associated with discrete representations. However, given that visual data are inherently continuous, most approaches require an additional discretization step, which, as discussed earlier, is far from straightforward. Recent studies (Li et al., 2024c) argue that the essence of autoregressive models lies in "next element prediction", regardless of whether the elements are discrete or continuous. This perspective revives the debate over the merits of *"discrete vs. continuous"* representations in autoregressive models. Continuous representations offer advantages, particularly in simplifying the training of visual data compressors. However, they also pose new challenges for designing autoregressive architectures. Firstly, the cross-entropy loss is no longer applicable. Although L2 loss is a simple alternative, it often compromises the quality and diversity of generated outputs (Li et al., 2024c). Thus, developing alternative loss functions tailored for continuous settings is still under exploration. Secondly, the challenge of multimodal adaptability arises. Large language models (LLMs) based on discrete representations have already shown remarkable capabilities, but integrating visual continuous and discrete language representations within a unified autoregressive framework remains a complex task. TransFusion (Zhou et al., 2024) has made pioneering efforts in this area, yet significant work remains to build upon these initial efforts. Currently, continuous visual representations have yet to demonstrate a decisive advantage over discrete ones in autoregressive

modeling. The full potential of autoregressive model-based continuous representations remains to be realized and represents a promising direction for future research.

**Inductive bias in autoregressive model architecture.** A key consideration in visual autoregressive modeling is whether a vanilla autoregressive model, without any inductive bias, truly represents the most optimal approach. Recent works (Sun et al., 2024a; Team, 2024) have shown that even standard language models like Llama (Touvron et al., 2023a;b) can generate high-quality images and achieve effective multimodal without requiring specific inductive bias. Nonetheless, there is still merit in exploring architectures that incorporate inductive biases tailored to visual signals. VAR (Tian et al., 2024) utilizes hierarchical multi-scale tokenization to capture visual structures better, while other approaches (Chang et al., 2022; Xie et al., 2024b) leverage masked image modeling strategies to enhance performance. Additionally, recent works (Zhou et al., 2024; Xie et al., 2024b) highlight the advantages of employing bidirectional attention over unidirectional ones for processing visual signals. Exploring how inductive biases can be integrated into autoregressive models, and their potential impacts on scalability and multimodal fusion, presents a challenging yet promising area for future research.

**Downstream task.** Current research on downstream tasks for visual autoregressive models largely focuses on zero-shot capabilities in a limited range of tasks or on designing models specifically tailored for individual downstream applications. In this respect, visual autoregressive models lag behind diffusion and language models. A critical open question remains whether visual autoregressive models can follow the trajectory of language models, which quickly adapt to new tasks via techniques such as prompt tuning and instruction tuning, or attain the versatility of diffusion models, which excel across various visual tasks including structure control, style transfer, and image editing. As research on autoregressive models advances, developing a unified autoregressive model that could adapt to downstream tasks is a primary focus in future work.

## 5.2 Application Roadmaps

**Long Video Generation** The generation of long videos remains an unsolved problem for various generative models. Due to the inherent limitations of current generation algorithms, the most advanced flow-based video generation models (OpenAI, 2024; Kuaishou, 2024) typically produce videos with durations similar to the training data, usually around 5 to 10 seconds. However, autoregressive models, particularly in the context of large language models (LLMs), have demonstrated strong extrapolation abilities, generating text sequences that far exceed the length of the sequences seen during training. This capability suggests that visual autoregressive models hold significant potential for generating long video sequences. Recent work (Jin et al., 2024b) has begun to explore the use of autoregressive models for generating longer video sequences, opening up new avenues for this challenging task.

**World Simulator for Embodied AI** A promising direction for the future of visual autoregressive generation is the development of world simulators that can facilitate the training of models in dynamic, interactive environments. These simulators would enable models to generate realistic visual content by interacting with simulated worlds in a context-aware manner. By integrating physical dynamics, environmental factors, and agent interactions, such simulators could allow visual autoregressive models to generate long and temporally consistent video sequences, building on the model's general ability to extrapolate from sufficient training data. Existing works (Hu et al., 2023; Bruce et al., 2024; Wu et al., 2024b; Cheang et al., 2024) have already made significant progress in leveraging autoregressive models as world models. The potential to scale training data and model size to enable the learning of general intelligence, allowing models to interact with the world and provide multimodal feedback, presents an exciting avenue for further exploration.

**Unified Multimodal Generation.** The future of multimodal autoregressive generation lies in advancing models capable of seamlessly combining and generating content across different modalities,

such as text, images, audio, and video. Autoregressive models naturally lend themselves to cross-modal generation, as they treat different modalities equally in the token space. While existing work has explored the potential of autoregressive models for multimodal generation (Lu et al., 2022; Wang et al., 2024d) and unified models for both understanding and visual generation (Team, 2024; Zhou et al., 2024; Xie et al., 2024b; Wu et al., 2024a), these models still fall short of the state-of-the-art models in each individual modality. The key challenge likely lies in the development of large-scale multimodal datasets and unified cross-modal representations. We hope an aha moment that, with sufficient training, autoregressive models will emerge with powerful cross-modal generation capabilities, where the strengths of one modality can boost the generation of others.

## 6 Conclusion

This survey provides a comprehensive overview of the autoregressive modeling landscape in computer vision, drawing parallels to its foundational success in NLP while highlighting the unique representational strategies inherent to visual data. By categorizing autoregressive models into pixel-based, token-based, and scale-based approaches, we offer a structured understanding of their underlying mechanisms. Furthermore, we illuminate the diverse applications of these models across domains such as image and video generation, multi-modal tasks, and medical applications. Finally, this survey addresses current challenges in visual autoregressive models, offering valuable insights into potential avenues for advancement in this rapidly evolving field.

## 7 Acknowledgement

This work was primarily supported by the Theme-based Research Scheme (TRS) project T45-701/22-R of the Research Grants Council (RGC), Hong Kong SAR, and partially supported by ONR, NSF, the Simons Foundation, as well as gifts and awards from Google, Amazon, and Apple.

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
