# OpenReview forum: "Autoregressive Models in Vision: A Survey"
_TMLR — Accepted by TMLR_

### Review · Reviewer_Fc6G · 2024-12-27

**Summary Of Contributions:**

The major contribution of this work is the comprehensive survey on autoregressive models for vision tasks. This survey covers methods, tasks and evaluation in vision tasks with a particular focus on autoregressive models. The methodologically, the models are categorized as three, pixel-based, token-based and scale-based. The tasks are grouped by image generation, video generation, 3D generation and multimodal generation.

**Audience:**

Yes

**Broader Impact Concerns:**

None primarily because it is a survey paper.

**Claims And Evidence:**

Yes

**Requested Changes:**

* Shorten section 3, since many are duplicated across several tasks.

* Clearly discuss the pros and cons of autoregressive models and its challenges when compared with non-autoregressive models.

* Similarly, differentiate evaluation methodologies which are specific to autoregressive modeling approaches.

**Strengths And Weaknesses:**

Strengths

* As noted in the contribution, it is a comprehensive survey on autoregressive models, covering diverse topics in the area. Grouping by methodologies and tasks sounds reasonable to me with much details with regards to the task specific models in Section 3.

Weaknesses

* Section 3 is lengthy in that most techniques are similar and potentially shared even in different tasks. I think this section could be shorten further by introducing the key techniques relevant to particular task settings.

* The pros and cons of autoregressive models when compared with other non-autoregressive models are not clear. Similarly, the challenges in Section 5 sound not specific to autoregressive models, but shared with other modeling approaches.

* Similarly, evaluation methods detailed in Section 4 do not sound quite specific to the autoregressive models, and it is better to differentiate what are unique to the autoregressive models or not.

---

> ### Author Response · Authors · 2025-02-16
> **Response to Reviewer Fc6G**
>
> > Improve the paper structure in Section 3.
>
> Thanks for your valuable suggestion\! We have streamlined the presentation in Section 3 and clarified the intentional design behind the cross-listing of certain methods, please see the general response for details.
>
> > Clearly discuss the pros and cons of autoregressive models and its challenges when compared with non-autoregressive models.
>
> Thanks for your valuable suggestions. We summarize the unique advantages of autoregressive generative models in three key points:
>
> 1\. **Scaling Laws:** The success of the next-token prediction paradigm in NLP is largely attributed to well-established scaling laws. While the scaling laws for methods like Diffusion and GANs remain underexplored, autoregressive visual models have the potential to transfer the successful scaling experiences from NLP. This could enable efficient scaling in visual generation frameworks. (See relevant papers: LlamaGen\[1\] and FLUID\[2\].)
> 2\. **Deployment Efficiency:** Autoregressive generative models can leverage existing deployment technologies designed for language models. For example, frameworks like VLLM provide autoregressive acceleration, significantly improving generation efficiency.
> 3\. **Bridging Language and Vision:** Autoregressive models represent a potential milestone in unifying multimodal understanding and generation. By closely aligning with the structures of language models, they may offer a seamless bridge between vision and language tasks.
>
> And the drawbacks can be summarized as:
>
> 1\. **Limitation in Generating Ultra-High-Resolution Images:** The autoregressive models typically have a higher-order time complexity(O(N^6)) compared to diffusion models when generating an NxN image.
> 2\. **Visual Quality:** While autoregressive generation has shown promising progress in prompt-following when generating images from given prompts, the visual quality of single images generated by autoregressive models still lags behind state-of-the-art diffusion or flow-based models.
>
> **We have supplemented a detailed discussion in Section 2.4.**
>
> > Clearly discuss the challenges specific to autoregressive models.
>
> Thanks. We have provided a more detailed discussion in Section 5. In addition to the technical challenges of autoregressive models covered in Section 5.1, which discusses different technical decisions and emerging architectural trends within autoregressive models, we have expanded Section 5.2 to include a forward-looking discussion on Future Application Roadmaps of autoregressive models, considering their unique advantages.
>
> > Differentiate evaluation methodologies which are specific to autoregressive modeling approaches.
>
> Thanks for your valuable suggestions. We believe that most evaluation metrics are more closely tied to the specific application tasks and the evaluation objectives, rather than the model architecture itself, as autoregressive methods typically require comparisons with non-autoregressive methods on these metrics. While we still agree that a more specific discussion of the metrics is necessary, we have expanded Section 4.3 to include a categorization of evaluation metrics based on task types. We hope that the revision will address your concerns.
>
> \[1\]. Sun, Peize, et al. "Autoregressive Model Beats Diffusion: Llama for Scalable Image Generation." arXiv preprint arXiv:2406.06525 (2024).
> \[2\]. Fan, Lijie, et al. "Fluid: Scaling autoregressive text-to-image generative models with continuous tokens." arXiv preprint arXiv:2410.13863 (2024).

---

> > ### Comment · Reviewer_Fc6G · 2025-03-03
> >
> > Thank you for the updates. I've checked the details and the revised one is clearly improved when compared with the initial draft.

---

### Review · Reviewer_TBJj · 2025-01-06

**Summary Of Contributions:**

This paper is a survey on auto-regressive models in computer vision, mostly focused on generative models. A very wide variety of methods is presented, divided into modalities: image, video, 3D, ect, but also into types of auto-regressive approaches: pixel-based, token-based and scale-based. Many illustrations help understand and categorize the different approaches. Finally, the paper proposes a discussion on challenges and future work for auto-regressive models.

**Audience:**

Yes

**Claims And Evidence:**

Yes

**Requested Changes:**

Requested changes:
- Fix the typos.
- Improve the paper structure to eliminates redundant presentation of the same method.
- Incorporate the suggested papers.

Questions and remarks:

- Are all token based approaches based on what is described in 2.2.2 ?

- “Pixel-based models (Sec. 2.2.1) generate images pixel by pixel, capturing the most fine-grained spatial details.” This is not necessarily always the case, self-supervised models such as DINO / DINOv2 are known to be extremely good for fine-grained tasks (segmentation for example), without reconstructing pixels.

- Regarding the length of the paper, I believe it is appropriate for a survey, which are generally much longer than regular papers.

**Strengths And Weaknesses:**

Strengths:

* The proposed survey is comprehensive with many relevant methods cited and presented. The methods are well organized into modalities, then tasks, then finally types of auto-regressive approaches. Overall, the survey is of good quality and will be useful for the community working on developing and scaling auto-regressive models.

* The survey is well illustrated and the presentation is clear. In particular, the tree diagrams are very informative and present a clear taxonomy of the methods. Additional figures explain well the different auto-regressive approaches.


Weaknesses:

* There is a “Video Tokenizer” paragraph in Section 3.1: Image Generation. Why not keep this for Section 3.2: Video Generation ? In result, MAGVIT and MAGVIT-v2 are discussed twiced in both 3.1 and 3.2, which makes the organization confusing. Also, both MAGVIT papers appear under “Unconditional Image Generation” in Figure 2, which is confusing as these are primarily designed to generate video.

* Make-A-Scene is discussed both in “3.1.2.a Token-wise Generation” and in “3.3.3 Scene Generation”, which similarly to MAGVIT makes the structure of the paper confusing.

* The survey is very much focused on generative models and misses on some interesting papers that are more focused on learning representations and world models rather than generating pixels:

         * GAIA-1: A Generative World Model for Autonomous Driving, https://arxiv.org/abs/2309.17080. This paper presents in my opinion one of the most interesting auto-regressive world models. The model is trained on unlabelled driving data and can predict many time-consistent frames into the future starting from a context.

         * The AIM serie: Scalable Pre-training of Large Autoregressive Image Models, https://arxiv.org/abs/2401.08541,  Multimodal Autoregressive Pre-training of Large Vision Encoders, https://arxiv.org/abs/2411.14402. These papers demonstrate the power of self-supervised learning and auto-regressive models to learn strong visual representations.



* Three auto-regressive categories are presented, pixel-based, token-based and scale-based. What about auto-regressive approaches in the embedding space, without discretization, are there such methods ? Also, is it worth having a “scale-based” category for a category for so few methods ? And in Figure 2, why only the “Unconditional Image Generation” and “Text-to-Image Synthesis” tasks have the categorization pixel/token/scale ?

* I have spotted several typos:
Sec 2.3 first line of the text -> “Variational Autoencoders a(VAEs)”
Page 11, paragraph 2, weird formulation -> “Zhu et al. (2024b) presents VQGAN-LC.”
Bottom of page 13 -> “RQ-VAE (?).”

---

> ### Author Response · Authors · 2025-02-16
> **Response to Reviewer TBJj**
>
> > Improve the paper structure in Section 3.
>
> Thanks for your valuable suggestion\! We have streamlined the presentation in Section 3 and clarified the intentional design behind the cross-listing of certain methods, please see the general response for details.
>
> > Incorporate the suggested papers.
>
> Thanks for pointing out this\! We agree that discussions about representation learning works based on autoregressive models are important. In the revision, we have added discussions and references to missing papers, including the introduction of GAIA-1\[1\] in Section 3.2.3 and the AIM series\[2\]\[3\] in Section 3.4.1.
>
> > Categorization Issues.
>
> **(1) Can non-discrete autoregressive methods be included in categorization？**
>
> We thank the reviewer for raising this important point. We have observed recent explorations of continuous autoregressive models in the field(e.g., Li et al., 2024\[4\], Zhou et al., 2024\[5\]), which are also included in our survey. We argue that our taxonomy is based on the representation strategy of the autoregressive elements, which naturally accommodates both discrete and continuous approaches. Although continuous autoregressive methods differ slightly—in that they operate directly in the embedding space and use alternative loss formulations—the core principles are analogous to those of discrete methods. We have added a discussion in Sec. 2.2 to clarify this distinction.
>
> **(2) Is it worth having a “scale-based” category?**
>
> We argue that it is necessary to categorize scale-based methods as a distinct paradigm. (1) Scale-based methods fundamentally differ from pixel-based and token-based approaches in that they treat specific resolutions as sequence elements, generating all content at the current resolution in each iteration. This distinction sets them apart from existing methods and justifies them as a separate category. (2) Scale-based methods offer unique advantages, such as reduced computational complexity (see Section 2.2.4) and promising scaling laws, which position them well to overcome the limitations of current paradigms and drive the growth of autoregressive generation within this new framework. Although there has been limited follow-up works due to the relatively recent introduction, ongoing research continues to expand in this direction. We believe in its potential and insist that it warrants recognition as a separate autoregressive generation paradigm.
>
> **(3) why only the “Unconditional Image Generation” and “Text-to-Image Synthesis” tasks have the categorization pixel/token/scale?**
>
> We adopt this because unconditional image generation and text-to-image generation are more foundational tasks that encompass a large portion works of visual autoregressive modeling, making them natural proxy tasks for studying autoregressive models. Related research typically focuses on improving visual generation and incorporating additional conditions, which can be extended to other applications. In these two subsections, we categorize methods by generation paradigms to highlight their contributions to the respective paradigms. In contrast, tasks like image editing are more defined by the task itself than by autoregressive generation, and therefore warrant a different treatment.
>
> > Minor Explanation of “Pixel-based models”
>
> Thank you for pointing out this\! Our original intent was to highlight that pixel-based models inherently capture fine-grained spatial details through their pixel-by-pixel generation. However, we acknowledge that token-based models and scale-based models can also excel in fine-grained tasks without reconstructing every pixel. We have revised the manuscript to clarify this point.
>
> > Other typos
>
> Thanks for your careful review\! We have corrected all the typos pointed out, as well as other errors we identified during revision.
>
> \[1\] Hu, Anthony, et al. "Gaia-1: A generative world model for autonomous driving." arXiv preprint arXiv:2309.17080 (2023).
> \[2\] El-Nouby, Alaaeldin, et al. "Scalable pre-training of large autoregressive image models." arXiv preprint arXiv:2401.08541 (2024).
> \[3\] Fini, Enrico, et al. "Multimodal autoregressive pre-training of large vision encoders." arXiv preprint arXiv:2411.14402 (2024).
> \[4\] Li, Tianhong, et al. "Autoregressive image generation without vector quantization." Advances in Neural Information Processing Systems 37 (2025): 56424-56445.
> \[5\] Zhou, Chunting, et al. "Transfusion: Predict the next token and diffuse images with one multi-modal model." arXiv preprint arXiv:2408.11039 (2024).

---

> > ### Author Response · Authors · 2025-03-10
> > **Response to Reviewer TBJj**
> >
> > Dear Reviewer,
> >
> > Thank you for your valuable feedback and thoughtful comments on our manuscript. We have carefully replied all the points raised in your review and hope that our revisions and responses have resolved your concerns. Please let us know if there are any additional points or clarifications needed, we are happy to provide further details.
> >
> > We greatly appreciate your time and effort in reviewing our work and look forward to hearing from you!

---

### Review · Reviewer_N3GT · 2025-02-02

**Summary Of Contributions:**

This survey provides a comprehensive review of autoregressive models in computer vision, with three major contributions:
1. A systematic categorization of visual autoregressive models into pixel-based, token-based, and scale-based approaches, with detailed technical analysis of each framework
2. A thorough examination of applications across multiple domains including image generation, video synthesis, 3D generation, and multimodal tasks, supported by ~250 references
3. A comparative analysis of existing models through standardized benchmarks and evaluation metrics

**Audience:**

Yes

**Claims And Evidence:**

Yes

**Requested Changes:**

Critical changes:
1. Add quantitative analysis of computational costs and efficiency trade-offs across different frameworks
2. Expand evaluation metrics discussion beyond standard benchmarks
3. Include more concrete future research directions with specific technical challenges

Beneficial changes:
1. Expand discussion on scaling behavior and resource requirements
2. Add comparisons of training stability across different approaches

**Strengths And Weaknesses:**

Strengths:
- Clear organizational structure that helps readers understand the progression of autoregressive models in vision
- Strong technical depth with detailed mathematical formulations and architectural explanations
- Comprehensive coverage of recent developments up to 2024
- Quantitative comparisons across different approaches through benchmark results
- Thorough analysis of relationships with other generative approaches (GANs, Diffusion, VAEs)

Weaknesses:
- Limited analysis of computational complexity trade-offs between different approaches
- Evaluation metrics focus mainly on standard benchmarks without sufficient discussion of task-specific metrics
- Discussion of future challenges and research directions could be more concrete

---

> ### Author Response · Authors · 2025-02-16
> **Response to Reviewer N3GT**
>
> > Add quantitative analysis of computational costs and efficiency trade-offs across different frameworks
>
> Thanks for the valuable suggestion. **We add a dedicated subsection in Section 2.2.4 to quantitatively compare the computational complexity and efficiency among three autoregressive paradigms.** Considering the task of generating an $N \times N$ image using a standard self-attention Transformer and an optional CNN-based tokenizer, we summarize the efficiency comparison of different autoregressive generation paradigms as follows:
>
> | Method                  | Require Tokenizer | Compression Ratio | Complexity                              | Efficiency    |
> |-------------------------|------------------|------------------|----------------------------------------|--------------|
> | Next-Pixel Prediction  |  ✘               | \-                | $O\_T(N^6)$                             | ✩          |
> | Next-Token Prediction  | ✔               | $k$              | $O\_T(N^6/k^6) \+ O\_C(N^2)$              | ✩✩         |
> | Next-Scale Prediction  | ✔               | $k$              | $O\_T(N^4/k^4) \+ O\_C(N^2)$              | ✩✩✩       |
>
> > Expand evaluation metrics discussion beyond standard benchmarks
>
> Thanks for the valuable suggestions. **In response, we have expanded the discussion in Section 4.3 of our manuscript to include a more detailed classification of evaluation metrics based on specific tasks.** In addition to categorizing the metrics previously introduced, we have also incorporated task-specific metrics and benchmarks. For example, we now include codebook utilization for evaluating tokenizer reconstruction, T2I-CompBench\[1\] for compositional generation tasks, and VBench\[2\], EvalCrafter\[3\] for text-to-video generation.
>
> > Include more concrete future research directions with specific technical challenges
>
> Thank you for your suggestion. Since we have already discussed the technical challenges in Section 5.1, **we expand Section 5.2 to include a forward-looking discussion on Future Application Roadmaps.** This section now covers three key topics: Long Video Generation, World Simulator for Embodied AI, and Unified Multimodal Generation.
>
> > Discussion about scaling behavior and training ability
>
> We appreciate your comments. **We supply the discussion about scaling behavior and training stability in Section 2.4.** Autoregressive models typically exhibit strong scaling behavior, both with respect to dataset size and model size. Given that autoregressive models are directly trained by minimizing the negative log-likelihood (NLL), they tend to demonstrate robust training performance compared to other generative models.
>
> \[1\]. Huang, Kaiyi, et al. "T2i-compbench: A comprehensive benchmark for open-world compositional text-to-image generation." Advances in Neural Information Processing Systems 36 (2023): 78723-78747.
> \[2\]. Huang, Ziqi, et al. "Vbench: Comprehensive benchmark suite for video generative models." Proceedings of the IEEE/CVF Conference on Computer Vision and Pattern Recognition. 2024\.
> \[3\]. Liu, Yaofang, et al. "Evalcrafter: Benchmarking and evaluating large video generation models." Proceedings of the IEEE/CVF Conference on Computer Vision and Pattern Recognition. 2024\.

---

> > ### Comment · Reviewer_N3GT · 2025-03-03
> > **Thank you for your response**
> >
> > Thank you for your response. I have checked the revised manuscript, and all my comments have been addressed.

---

### Author Response · Authors · 2025-02-16
**General Response**

We would like to express our sincere gratitude to all the reviewers for their time and constructive feedback. We have revised our manuscript and submitted an updated version for review. **The revisions are clearly highlighted in blue for easy reference.**



## Summary of Revision

### Section 2

1. We add a comprehensive discussion on continuous autoregressove methods and address the question of whether these can be integrated into our categorization. The answer is yes (Reviewer TBJj);
2. We revise the statement about next-pixel prediction paradigm (Reviewer TBJj);
3. We include an in-depth analysis about the computational costs among three autoregressive paradigms(Reviewer N3GT);
4. We add a new discussion about the pros and cons of autoregressive models compared with non-autoregressive models(Reviewer N3GT Reviewer Fc6G);

### Section 3

1. We have streamlined the presentation by removing redundant descriptions of the same methods, such as Make-a-scene (Reviewer TBJj, Reviewer Fc6G);
2. We add a new introduction to works focus on representation learning in generation tasks, including GAIA-1\[1\], AIM\[2\], and AIMV2\[3\](Reviewer TBJj);
3. We move the discussion about video tokenizer from Section 3.1.1 to Section 3.2.1(Reviewer TBJj);

### Section 4

1. In Section 4.3, we include a new subsection to better categorize task-specific evaluation metrics.

### Section 5

1. We expand Sectopm 5.2 to include a forward-looking discussion on Future Application Roadmaps(Reviewer N3GT), including:
   1. Long Video Generation;
   2. World Simulator for Embodied AI;
   3. Unified Multimodal Generation;

## Structural Redundancy in Section 3

Reviewer TBJj and Reviewer Fc6G mentioned that some works are duplicated across several tasks in Section 3, which we believe will be a concern for all reviewers. Below we clarify our revisions and design rationable:

**(1) Revisions Implemented:**

1. **Relocation of Video Tokenizer Content:** We have moved the “Video Tokenizer” paragraph to Section 3.2.1 to enhance the clarity and completeness of our discussion on video generation.
2. **Removal of Redundant Content:** We have eliminated duplicated discussions (e.g., the repeated statement of Make-a-Scene in Section 3.3.3) to streamline the section.
3. **Additional Discussion:** We have expanded discussions for certain works to provide clearer insights into their contributions.

**(2) Design Rationable:**

The cross-listing of seminal works across Section 3.1.1 and Section 3.2.1 reflects a deliberate design choice rather than oversight. We argue that these works contribute inherently advance two frontiers: 1\) image/video tokenization, and 2\) autoregressive sequence modeling strategies. Our aim is to emphasize these distinct aspects without redundancy. This structure allows readers who are interested in a specific component—such as image tokenizers—to quickly locate the relevant information.

\[1\] Hu, Anthony, et al. "Gaia-1: A generative world model for autonomous driving." arXiv preprint arXiv:2309.17080 (2023).
\[2\] El-Nouby, Alaaeldin, et al. "Scalable pre-training of large autoregressive image models." arXiv preprint arXiv:2401.08541 (2024).
\[3\] Fini, Enrico, et al. "Multimodal autoregressive pre-training of large vision encoders." arXiv preprint arXiv:2411.14402 (2024).

---

### Decision · Action_Editor_Y2T5 · 2025-03-07

**Recommendation:** Accept as is

**Comment:**

All reviewers agree that the survey is a valuable contribution to the community. It is timely, well-written, and well-illustrated.

**Audience:**

The paper should have a significant audience as it serves both as an overview and as an introduction to a large field of research.

**Claims And Evidence:**

The paper surveys recent developments in autoregressive models. The descriptions are accurate and well-communicated. The presented dichotomy is both reasonable and didactic.